# Decoding the Different Aroma-Active Compounds in Soy Sauce for Cold Dishes via a Multiple Sensory Evaluation and Instrumental Analysis

**DOI:** 10.3390/foods12193693

**Published:** 2023-10-08

**Authors:** Dandan Pu, Yige Shi, Ruixin Meng, Qianqian Yong, Zongyi Shi, Dandan Shao, Baoguo Sun, Yuyu Zhang

**Affiliations:** 1Food Laboratory of Zhongyuan, Beijing Technology and Business University, Beijing 100048, China; 18518351472@163.com (D.P.); jennyshiyige@163.com (Y.S.); mengrx0701@163.com (R.M.); sunbg@btbu.edu.cn (B.S.); 2Key Laboratory of Flavor Science of China General Chamber of Commerce, Beijing Technology and Business University, Beijing 100048, China; 3Key Laboratory of Geriatric Nutrition and Health, Beijing Technology and Business University, Ministry of Education, Beijing 100048, China; 4Yantai Shinho Enterprise FOODS Co., Ltd., Yantai 264000, China; yongqianqian@shinho.net.cn (Q.Y.); shizongyi@shinho.net.cn (Z.S.); shaodandan@shinho.net.cn (D.S.)

**Keywords:** aroma-active compound, check-all-that-apply, gas chromatography-mass spectrometry/olfactory, quantitative descriptive analysis, soy sauce

## Abstract

Screening the suitability of soy sauce for specific cooking methods from various products is beneficial for the fine development of the soy sauce industry. Multiple sensory evaluation and gas chromatography-mass spectrometry/olfactometry (GC-MS/O) analysis were combined to decode the suitability of soy sauces for cold dishes and characterize their differential aroma-active compounds. Thirty-two kinds of soy sauce with 42 sensory descriptors were determined via a check-all-that-apply analysis, and werefurther classified into six categories via a cluster analysis. The sensory evaluation results showed that seven soy sauce samples had the highest acceptance in each category. Solid-phase microextraction and solid phase extraction results combined with the GC-MS/O analysis results showed that a total of 38 aroma-active compounds were identified in seven soy sauce samples, among which 2-methoxy-phenol (6–93), ethyl acetate (2–48), 3-methyl-1-butanol (4–30), 3-methyl-butanal (5–24), methional (0–22), dimethyl trisulfide (5–19) and dimethyl disulfide (0–8) showed a higher relative odor activity value (ROAV). A partial least squares regression prediction combined with additional tests further confirmed that 2,5-dimethyl-pyrazine; 2,6-dimethyl-pyrazine; and 2-ethyl-6-methyl-pyrazine significantly contributed to the roasted attributes, methional significantly contributed to the sauce-like notes, ethanol significantly contributed to the alcoholic notes and 2-methoxy-phenol significantly contributed to the smoky notes. 2,5-Dimethyl-pyrazine; methional; 2,6-dimethyl-pyrazine and 2-ethyl-6-methyl-pyrazine significantly contributed to the caramel-like attributes.

## 1. Introduction

Soy sauce, a traditional Chinese condiment brewed with beans, wheat, bran, salt, and a starter with a unique flavor, can improve the aroma, taste, and color of dishes, as well as promoting our appetite. Generally, soybeans/soybean flakes provide the main protein source, wheat/wheat flour provides the main carbohydrate source, and the starter mainly provides microorganisms such as Aspergillus oryzae/Aspergillus sojae, salt-tolerant yeast, lactic acid bacteria, etc. [1]. Soy sauce can be divided into cooking soy sauce and table soy sauce according to consumers’ eating habits. Cooking soy sauce is suitable for cooking and heating but it is best to not eat it directly [2]. Soy sauce that is suitable for cold food can be used to make cold dishes and served with sushi and sashimi.

In 2021, the total output of soy sauce in China reached about 7.78 million tons, accounting for more than 89.65% of the global output. The aroma of soy sauce has been studied for more than 100 years [3], and more than 300 aroma compounds have been identified among different soy sauce samples, including alcohols, aldehydes, esters, acids, pyrazine, furans, ketones, phenols, alkanes, etc. [4,5,6,7]. These compounds, with alcoholic, malty, caramel-like, smoky, flowery and fruity characteristics are forming the basic aroma profiles of soy sauce [6,8]. The key aroma compounds of soy sauce that have been reported so far are methional; guaiacol; 2,5-dimethyl-4-hydroxy-3(2H)furanone (HDMF); 2-ethyl-4-hydroxy-5-methyl-3(2H)furanone (HEMF); guaiacol; 4-ethyl guaiacol; 2/3-methylbutanal; etc. [9,10].

Different kinds of soy sauce are suitable for different dishes due to their variance in aroma characteristics. Choosing the right soy sauce can make it possible to cook dishes with better flavor. Therefore, developing efficient and precise screening methods is of great significance for the precise development and quality improvement of soy sauce. Check-all-that-apply (CATA) questions allow consumers to select all potential attributes from a given list to describe a test product to collect information about perceived product attributes [11]. CATA can display the attribute information of samples that is detected by consumers, as well as the relationship between these attributes and the general preference or acceptance. This method can directly identify the preference drivers of consumer groups with different preference patterns. With few instructions, it is relatively easy to implement, making it possible to quickly understand the sensory attributes of consumers for different goods [12]. More recently, CATA questions have been used to classify consumers’ flavor preferences [13]. Lin et al. [14] reported that the aroma composition sensory properties of Chinese bog bilberry wines were characterized using CATA and GC-Quadrupole-MS analysis. At present, solid-phase microextraction (SPME) combined with gas chromatography-mass spectrometry (GC-MS) analysis is the most efficient method to identify the aroma compounds in soy sauce [15,16]. SPME and stir bar sorptive extraction are the most common aroma enrichment methods for volatile extraction from soy sauce in recent years due to the lack of consumption of organic solvents [6,17]. Additionally, they are fast, easy to operate, and cost-effective.

Additionally, to correlate the relationship between the sensory properties and chemical data, many multiple analysis methodshave been performed. For example, Li et al. [18] investigated the variances among the different types, processes, and regions of soy sauce via a principal component analysis (PCA), hierarchical cluster analysis (HAC) and orthogonal partial least squares discriminant analysis (OPLS-DA). Diez-Simon et al. [19] comprehensively characterized the volatile compounds and related compositional differences in the aroma profiles to the origin and production history of the samples via PCA, heatmap analysis, and partial least squares-discriminant analysis (PLS-DA).

In this work, CATA analysis was performed to detect the aroma profile variances among different brands of soy sauce and to divide them into different clusters. Then, the application scenario of soy sauce in cold dishes was simulated to select the soy sauce that is suitable for cold dressing via a preference ranking analysis. The aroma characteristics of the favored soy sauce were determined via a quantitative descriptive analysis (QDA) to compare their differences. SPME-GC-MS and GC-MS/Olfactory (GC-MS/O) were combined to analyze the aroma-active compounds of the selected soy sauce. Then, a partial least squares regulation analysis (PLSR) was performed to correlate the relationship between the aroma-active compounds to the sensory attributes and further clarify their contributions. Finally, their contributions were further confirmed via additional tests. This work proposes an effective method for the rapid screening of applicable products, which help to select practical application scenarios and improve the quality of soy sauce in the future.

## 2. Materials and Methods

### 2.1. Materials and Chemicals

A total of 32 different kinds of soy sauce, coded as XH1, XH2, XH3, XH4, XH5, XH6, XH7, XH8, YZ, HT1, HT2, HT3, HT4, JJ1, LJJ2, LJJ3, LJJ4, LJJ5, CB1, CB2, CB3, CB4, JJ1, JJ2, LCC1, LCC2, LCC3, BT1, BT2, HJ, WZ, and MG were used in this work. All samples were purchased from a local supermarket. All samples were stored in their original containers and sealed at room temperature (25 °C) prior to analyzsise. 

Chromatographic grade methanol (99.9%) was purchased from Merck Chemical Company (Shanghai, China). 2-Methyl-3-heptanone (99.9%), and 5-ethyl-4-hydroxy-2-methyl-3(2H)furanone (99.9%) were purchased from Macklin Biochemical Company (Shanghai, China). 2,5-Dimethyl-4hydroxy-3(2H)-furanone (99.9%) was purchased from Sigma-Aldrich (Steinheim, Germany). Ethyl acetate (99.9%), 4-ethyl-2-methoxyphenol (99.9%), and 3-methylbutanal (99.9%) were purchased from J&K Scientific (Beijing, China). C_6_–C_28_ n-alkanes standard mixtures (≥97%) were purchased from Sigma Aldrich (Steinheim, Germany). Purified water was purchased from Hangzhou Wahaha Co., Ltd., Hangzhou, China.

### 2.2. Sensory Evaluation

The aroma profile analysis of the soy sauce samples involved evaluations by 12 trained panelists (6 males and 6 females). Twelve sensory evaluation panelists aged 24–30 and with no rhinitis were recruited from our laboratory. All panelists had sensory evaluation experiences in meat products and flavorings, and were also familiar with the quantitative descriptive analysis (QDA) evaluation methods. The panelists were trained to discriminate the aroma differences of 54-aroma kits (Le Nez du Vin, France) for three weeks before sensorily evaluating the soy sauce samples, according to our previous work [20,21]. The aroma descriptions of the soy sauce sample were analyzed using CATA analysis [22]. The soy sauce (20 mL), loaded into odorless transparent plastic bottles and coded with 3-digit numbers was submitted to the panelists randomly. The temperature and humidity of the sensory evaluation room were 24 °C and 55%, respectively. The procedures of the soy sauce aroma profile evaluation via CATA analysis was as follows: (1) Collecting and summarizing the aroma descriptions from the literatures and the descriptors perceived by the panelists [5,6,7,16,17]. (2) Screening the aroma descriptors of the soy sauces; 43 non repetitive sensory descriptors were selected to form a aroma vocabulary library for the soy sauces. (3) Each panelist was requested to check the perceived aroma descriptors for each kinds of soy sauce during evaluation. A cluster analysis was conducted to classify the different clusters of the 32 soy sauce samples based on the results of the CATA analysis. Then, the panelists were request to selected the best most acceptable acceptance samples from each cluster according to consumers’ preferences [23,24].

For the QDA analysis, the soy sauce samples were based on the results of the consumer preference test. Based on the frequency statistics of the aroma descriptors and the discussion of the panel group, 8 aroma attributes of soy sauce were determined: savory, caramel-like, smoky, cooked potato-like, malty, alcoholic, fruity, and sour. The amount of soy sauce added to the different dishes was summarized according to the book A Bite of China (Figure 1A). To simulate the simple cold dish condition without the consideration of the interactions between the different kinds of dishes and soy sauce samples, a soy sauce solution was created by adding soy sauce into distilled water based on the statistical results (Figure 1A). The soy sauce solution was loaded into an odorless transparent plastic bottle and coded with 3 digits randomly, and was subsequently submitted to the panelists. The panelists were required to score the intensity of 8 aroma attributes (1–3, weak;4–6, medium; 7–9, strong). All samples were repeated in triplicates.

### 2.3. Solid-Phase Micro Extraction (SPME)

The soy sauce samples for the SPME analysis were the same as those for the QDA analysis; 4 mL of soy sauce was loaded into a 20 mL SPME glass bottle and 0.7 g of NaCl was added to saturate the sample. The loaded soy sauce sample was heated at 45 °C for 20 min, and then extracted at 45 °C for 40 min with 75 μm of CAR/PDMS fiber (Supelco, Steinheim, Germany). Before volatile extraction the fiber was heated at 250 °C for 10 min. All the extraction procedures were conducted with the automatic headspace sampling system (CTC, Switzerland). Ten microliters of 2-methyl-3-heptanone (10.00 mg/L, dissolved in methanol) was added into the SPME vial as per the internal standard. All samples were repeated in triplicates.

### 2.4. Gas Chromatography-Mass Spectrometry (GC-MS) Analysis

An Agilent 8890 GC instrument equipped with a 5977B single quadrupole mass spectrometer (GC-MS) was used in this work. The soy sauce aroma compounds were separated using a DB-Wax column (30 m × 0.25 mm, id. 0.25 μm). Helium (purity of 99.999%) was used as the carrier gas, and the flow rate was 1.55 mL/min. The pulsed splitless mode was used during the SPME fiber desorption at the injection port (250 °C) for 5 min. The column oven was programmed starting at 35 °C for 1 min, and was increased to 100 °C at a rate of 4 °C/min and held for 1 min, then increased to 170 °C at a rate of 2 °C/min and held for 1 min, and increased again to 220 °C at a rate of 5 °C/min before, finally being held at 220 °C for 1 min. The ionization mode of the mass spectrometry condition was the electron impact (EI) mode; the electron energy was 70 eV; the ion source temperature was 230 °C; the quadrupole temperature was 150 °C. The full scan mode was used qualitatively, and the mass scanning range was 3–450 *m/z*. Methional (48, 76, 104) was quantified using the selected-ion-monitoring (SIM) mode.

### 2.5. GC-MS/Olfactometry (GC-MS/O) Analysis

To screen the aroma-active compounds in different brands of soy sauce (the sauce samples for the GC-MS/O analysis were the same as those for the SPME-GC-MS analysis), the olfactory detection port (ODP4, Gerstel, Germany) and MS with a ratio of 1:1 were combined after the separation outlet of the column. Volatile extracts for the GC-MS/O analysis were obtained by using SPME, employing the sampling procedure described in Section 2.3. The capillary column (0.1 mm) of the transfer line was kept under a constant temperature of 250 °C. The analytical parameters were the same as those for the GC-MS analysis. The sniffing procedure for each soy sauce was carried out by at least three individual panelists. A compound was recorded as being aroma-active when it was perceived or described using similar odor qualities by at least two panelists at the same retention time. The sensory attributes perceived by the panelists at the ODP were linked to specific compounds by matching the retention indexes and by validating these compounds with aroma-active compounds that have been previously reported in soy sauce.

### 2.6. Identification and Quantification of Aroma Compounds

The aroma identification based on the Kovats retention index (RI), the mass spectrum library (NIST 20), standard compounds (S), and odor quality (O). The RI values were calculated using a series of linear alkanes (C6–C28) according to a modified Kovats method [25]. 2-Methyl-3-heptanone was used as the internal standard to calculate the relative concentration of the volatile compounds detected in the soy sauce. The aroma compounds were quantified on the DB-WAX column, and their con-centrations were determined based on the ratio of the peak area of the odorant com-pound relative to the peak area of the internal standard against the ratio of their con-centrations. All samples were repeated in triplicates.

### 2.7. Statistical Analysis

All of the data were collected and statistically analyzed using Microsoft Excel 2016. The PLSR (the relationship between the aroma-active compounds and sensory attributes) was conducted using Xlstat 2016 (Addinsoft, New York, NY, USA). The QDA figures of the aroma profiles of the different soy sauce samples were performed using Origin Pro 2021 (OriginLab Corporation, Northampton, MA, USA).

## 3. Results

### 3.1. Comparison of Product Descriptions via CATA

A total of 42 aroma attributes (Figure 2), including savory, sauce-like aroma, milk-like, caramel-like, cheesy, buttery, cooked meat-like, fatty, roasty, smoky, garlicky, cooked onion-like, seasoning-like, etc., from the 32 kinds of soy sauce samples were perceived by the panelists during the CATA analysis. Among them, savory, sauce-like aroma, sour, smoky, and bean aroma showed the highest frequency in all soy sauce samples, whereas the fruity attributes such as cucumber, citrus, blueberry, grape, floral, etc., showed lower frequencies. Based on the aroma profile properties, the thirty-two soy sauce samples were divided into six categories: Category 1: the HJ, LJJ2, HT2, MG, and LJJ5 soy sauces; Category 2: the CB1, JJ1, XH5, XH3, CB4, and YZ soy sauces; Category 3: the XH1, XH8, WZ, HT4, XH4, and XH6 soy sauces; Category 4: the CB3, LCC2, HT3, HT1, and CB2 soy sauces; Category 5: the LJJ4, LCC1, LCC3, BT2, and LJJ3 soy sauces; and Category 6: the LJJ1, XH2, BT1, XH7, and JJ2 soy sauces. Among all soy sauce samples, the savory, sauce-like aroma, sour, smoky, and bean aroma had higher frequencies and greater correlation coefficients than the other soy sauce aroma attributes.

### 3.2. Consumer Preference and Aroma Descriptive Analysis of Different Soy Sauces

To obtain more accurate addition amounts of soy sauce into different dishes during cooking or cold dish preparation, a total of 433 recipes, including seven kinds of traditional Chinese dishes with different cooking methods (salad, stir-fry, stew, steamed, soup, grilled, and snacks), were summarized, and the amount of soy sauce added varies depending on the ingredients and cooking method. The addition amounts of soy sauce in different categories of dish are shown in Figure 1A. Among the different cooking methods, the amount of soy sauce added to cold dishes is the highest, followed by steamed dishes, because these two kinds of dishes mainly rely on soy sauce to promote the umami and savory flavor. The lowest soy sauce addition amount was for snack foods. The average soy sauce addition amount (1.74%, 0.348 g) for cooking simulations was finally determined. Subsequently, soy sauce preference evaluation tests were performed by adding 0.174 g of soy sauce into 50 mL of purified water.

The preference results (Figure 1B) showed that seven soy sauce samples, including XH5, HJ1, XH8, BT1, CB3, YZ, and LJJ4, had the highest rank sum during simulations for cold dishes. These seven soy sauce samples were all produced using the high-salt liquid fermentation method. The results of the preference evaluation showed that the YZ soy sauce was the most popular (with the strongest fruity aroma), followed by the XH8 soy sauce, the BT1 soy sauce, the XH5 soy sauce, the HJ1 soy sauce, the CB3 soy sauce, and the LJJ4 soy sauce.

QDA was subsequently performed to analyze the aroma profiles of these seven soy sauce samples. However, before the QDA analysis, the aroma profiles of these samples should be determined based on the aroma attributes of the CATA analysis. Due to the presence of many aroma attributes, more accurate and representative aroma profiles of each soy sauce should be determined. Eight aroma descriptions, including sauce-like, caramel-like, smoky, roasted potato, malty, sour, alcoholic, and fruity, were determined after a panel discussion. The QDA results are shown in Figure 1C; the screened-out soy sauces that were suitable for cold dishes had relatively strong intensities for the sauce-like and caramel-like attributes, but there were some differences in their aroma intensities. The XH8 soy sauce had a stronger sauce-like aroma and malty aroma. The YZ soy sauce had stronger-intensity caramel-like, smoky, and fruity notes. The BT1 soy sauce had a stronger sour aroma, while being weak in its smoky and caramel-like attributes. The CB3 soy sauce and LJJ4 soy sauce had weaker-intensity overall aroma profiles. The CB3 soy sauce had the lowest alcoholic, smoky, and sauce attributes. The LJJ4 soy sauce had the weakest intensity sour, roasted potato, and malty notes. The XH5 soy sauce showed a lower-intensity fruity aroma.

### 3.3. Comparison of the Aroma Composition of Different Types of Soy Sauce

The composition and content of the main aroma compounds among seven soy sauce samples were different. The mass spectrum-based non-targeted metabonomics workflow showed that a total of 101 volatiles were detected via SPME-GC-MS (the seven total ion chromatograms are shown in Appendix A), including esters (11), ketones (17), alcohols (15), furans and furanones (15), aldehydes (10), pyrazines (13), acids (9), phenols (5), and sulfur-containing compounds (6). Among them, 38 aromatic-active compounds were identified via GC-MS/O (Table 1), including five acids, two aldehydes, four furans and furanones, five alcohols, three phenols, six pyrazines, four sulfur-containing compounds, six ketones, and one other compound. It was found that seven kinds of soy sauce contained 25 common aromatic compounds. The content of the total volatile concentrations in each type of aroma compound is shown in Figure 3.

The alcoholic compounds (380.847–4132.622 μg/L) had the highest content in the soy sauces except for in the CB3 sample; this result might be due to differences in the fermentation processes, which are consistent with the QDA results. Ethanol showed the highest content among all alcohols, exhibiting the main contribution to the alcoholic aroma. The highest ethanol content in the XH8 soy sauce reached 4132.622 μg/L, and the minimum ethanol content in the CB3 soy sauce was 380.847 μg/L. The reason for the large difference in the ethanol content may be due to the different fermentation methods and raw materials of different types of soy sauce. The proportion of wheat in Japanese soy sauce is high; therefore, yeast could promote the fermentation of alcohol [18], resulting in a higher content of ethanol. Fermentation in southern China is more dependent on sunlight and environmental temperatures, and local microorganisms participate in the fermentation process, increasing the amount of volatile compounds. In addition, 1-octene-3-ol, produced via the lipid oxidation of aspergillus, is an aroma-active compound that was detected in the soy sauces [4,5]. Phenylethyl alcohol (24.891–125.174 μg/L), with a floral fragrance and rose-like odor, was one of the main compounds related to the floral attributes of soy sauce. It is only produced with yeast from phenylalanine during the traditional fermentation process [26]. Additionally, phenylacetaldehyde (1.676–26.602 μg/L), with sweet or honey characteristics, has been identified in many fermented soybean foods, and its relative concentration usually reaches the highest level at the late fermentation stage of soy sauce [27]. Among them, 3-methyl-1-propanol, 2,3-butanediol, and 2-ethyl-1-hexanol showed fruity and sweet characteristics, and benzyl alcohol and phenylethanol showed floral notes. In the long moromi fermentation stage under aerobic conditions, these compounds could be produced from sugars and branched-chain or aromatic amino acids [28].

The second abundant aroma compounds were the pyrazines (26.650–370.734 μg/L) with roasted and cocoa attributes. The content of pyrazine compounds in the CB3 soy sauce was the highest (370.734 μg/L). Pyrazines are produced through a slow Maillard reaction and heat sterilization during fermentation, and their concentrations are affected by the fermentation temperature and sterilization conditions [29]. Pyrazines are also important odorants in yeast extracts, which can be affected by the production conditions (temperature and water activity) of yeast extracts [30]. Therefore, pyrazine compounds in the soy sauce samples could also be migrated from the addition of raw materials.

Phenols are important compounds that contribute to smoky and sauce-like notes. Four phenol compounds, including 2-methoxy-phenol, 4-ethyl-2-methoxy-phenol, 4-ethyl-phenol, maltol, and ethyl maltol were detected via GC-MS/O. Among them, 2-methoxy-phenol (2.759–44.862 μg/L) and 4-ethyl-2-methoxy-phenol (5.648–119.76 μg/L), with burning, smoking, and spicy attributes, played an important role in the overall aroma of the soy sauces. 4-Ethyl-2-methoxy-phenol, with a significantly higher ROAV value ranging from one to seven, has also been confirmed as a key aroma compound in soy sauce [5,31]. These phenolic compounds are mostly formed from hydroxycinnamic acid and hydroxybenzoic acid production during koji fermentation [4]. Ferulic acid is transformed into 4-ethyl-2-methoxy-phenol via the decarboxylation of candida species rather than daunorchia species during fermentation [29]. Maltol and ethyl maltol, producing sweet and caramel-like aromas, are important aroma-active compounds in many foods. Maltol (1.970–17.212 μg/L), derived from sugar via 2,3-enolization during heating or directly from Amadori products, was detected in all samples [32]. It exists in steamed soybean, which is the raw material of most soy sauces [4]. Ethyl maltol, the flavor enhancer, was detected in the CB3 and LJJ4 soy sauces, and its relative concentration in the CB3 soy sauce was the highest (221.431 μg/L).

Esters related to the lipid metabolism of yeast are mainly accumulated in the middle stage of fermentation [26]. Ethyl esters are usually produced through the microorganism metabolism during the fermentation process; the generation of ethyl esters is extremely variable depending on fermentation conditions, such as the temperature or microorganism species [33]. High-molecular-weight fatty acid esters, including ethyl linoleate and ethyl oleate, are produced via the long-term thermostatic fermentation of long-chain fatty acids in the presence of fungal lipase [34]. However, they were not detected in the GC-O analysis, indicating their subtle contributions to the characteristic aroma of soy sauce due to their high threshold values.

3-Methyl-butanal with a malty note was detected in seven samples (5.353–26.279 μg/L). It is one of the important aroma-active compounds in soy sauce due to its high ROAV value (5–23). 3-Methyl-butanal could be generated from the amino acids isoleucine and leucine through Strecker (Maillard) reactions or microbial catabolism [35]. Moreover, isoleucine and leucine are metabolized in yeast through the Ehrlich pathway to produce 3-methylbutanal as an intermediate, which can be further oxidized to corresponding acids or reduced to corresponding alcohols [6]. 3-Methylbutanol also has a malty aroma, but its contribution to the overall aroma profiles was lower due to its far higher odor threshold than aldehyde compounds. Methional, with a cooked potato characteristic, was considered as the key odorant affecting the sauce-like and cooked potato attributes of the soy sauce. Methional is generated during Strecker degradation and fermentation [36], during heat sterilization or cooking. 3-(Methylthio)-1-propanol is produced in large quantities from the Strecker degradation and fermentation of methionine [37], which could increase the intensity of the roasted potato attribute [38]. It existed in low concentrations; however, it easily exceeded the threshold value of a positive nose odor [39]. In this experiment, SIM scanning was used to identify the co-outflow of 3-(methylthio)-1-propanal and acetic acid. Other sulfur compounds, including dimethyl disulfide and dimethyl trisulfide with cooked onion notes, could also be detected via GC-MS/O.

The sour property of the soy sauces was mainly caused by acids. Among all the acid compounds, the relative concentration of acetic acid was the highest (19.807–124.955 μg/L). Acetic acid is produced by lactic acid bacteria in the process of mori fermentation [29]. In addition, 3-methyl-butanoic acid (1.913–39.130 μg/L) with cheese or sweat notes is derived from the metabolism of amino acids in yeast through the oxidation of corresponding aldehyde intermediates [40], and it was detected in all samples. Short-chain fatty acids such as propanoic acid could originate from the lipolysis of microorganisms and provide cheese-like or rancid notes [19]. The sorbic acid that existed in the CB3 soy sauce was derived from potassium sorbate, because potassium sorbate is one of the preservatives for that soy sauce.

HDMF (0.074–1.176 μg/L) and HEMF (0.415–9.776 μg/L) are tautomers with a strong caramel-like aroma that were detected in all samples. These furanones are formed via Maillard reactions, but can also be produced through the microbial metabolism. It has been proved that they occur naturally in different yeast species, and are mainly related to the carbohydrate metabolism of Rouxi yeast [41]. 2,3-Pentanedione might be produced in the process of amino-acid-mediated chain elongation [42]. 2,3-Pentanedione was not detected in the XH5 soy sauce, but it was detected in other samples (0.691–9.012 μg/L).

### 3.4. The Relationship between Sensory Evaluation and the Aroma Compounds

In order to find out the contributions of volatile compounds to the aroma profiles of soy sauce, the sensory evaluation data and aroma compounds were correlated and analyzed through PLSR. PLSR analysis is applicable to cases with few samples but many independent variables [43,44]. The results showed (Figure 4A) that most X variables (the relative concentration of volatile compounds) and Y variables (the intensity of the sensory attributes) are loaded within the circle in the figure (r^2^ = 95%, r^2^ represents the degree of interpretation). Dimension 1 explained 24.90% of the predictive variables (major volatiles) and 46.10% of the responses (aroma perception), while dimension 2 explained 65.70% of the predictive variables and 62.80% of the response variance. The results showed that seven soy sauce samples could be divided into two groups according to dimension 1. The XH8, XH5, BT1, and YZ soy sauces were located on the positive axis of dimension 1, and LJJ4, HJ, and CB3 were located on the negative axis of dimension 1. YZ showed a strong aroma of roasted potato, while XH8 showed a strong malty aroma. XH5, XH8, LJJ4, BT1, and HJ were located on the positive axis of dimension 2, while YZ and CB3 were located on the negative axis of dimension 2. Among these 37 compounds, five of them had variable importance in projection (VIP) values of ≥ 1 among dimension 1, dimension 2, and dimension 3. However, more variables could be derived by adjusting the VIP value to 0.8 [45]. Therefore, 17 compounds with a VIP value greater than 0.8 and significant contributions to the soy sauce aromas (*p* < 0.05) were selected. Subsequently, based on the results of the standard correlation coefficients of the aroma compounds to the sensory attributes of soy sauce, the relationships between the potent aroma-active compounds and the sensory attributes were predicted, as shown in Figure 4B. The results showed that ethanol; 1-propanol; 2,3-pentanedione; methyl-pyrazine; 1-hydroxy-2-propanone; 2,5-dimethyl-pyrazine; 2,6-dimethyl-pyrazine; 2-ethyl-6-methyl-pyrazine; methional; propanoic acid; 2,3-butanediol; 3-methyl-butanoic acid; 4-methyl-pentanoic acid; 2-methoxy-phenol; maltol and 2,5-dimethyl-4-hydroxy-3(2H)furanone were positively correlated to the malty attribute. 1-Propanol; 2,3-pentanedione; methyl-pyrazine; 1-hydroxy-2-propanone; 2,5-dimethyl-pyrazine; 2,6-dimethyl-pyrazine; 2-ethyl-6-methyl-pyrazine; methional; propanoic acid; 2,3-butanediol; 3-methyl-butanoic acid; 4-methyl-pentanoic acid; 2-methoxy-phenol; maltol and 2,5-dimethyl-4-hydroxy-3(2H)furanone were positively correlated to the roasted potato attribute. Ethanol; 1-propanol; methyl-pyrazine; methional; propanoic acid and 2,3-butanediol were positively correlated to the sauce-like aroma. 1-Propanol; methyl-pyrazine; 2,6-dimethyl-pyrazine; 2-ethyl-6-methyl-pyrazine; methional; propanoic acid; 2,3-butanediol; 3-methyl-butanoic acid; 4-methyl-pentanoic acid; maltol and 2,5-dimethyl-4-hydroxy-3(2H)furanone were positively correlated to the caramel-like attribute. Ethanol; 1-propanol; methyl-pyrazine; methional; propanoic acid and 2,3-butanediol were positively correlated to the smoky attribute. Ethanol; 1-propanol; methional and 2,3-butanediol were positively correlated to the alcoholic attribute.

Additional tests were conducted to confirm the contribution of these aroma-active compounds to the aroma attributes. Seven odorants with high VIP values were selected and added into the sauce sample with the maximum relative concentration difference (Table 2). The additional tests results showed that 2,5-dimethyl-pyrazine; 2,6-dimethyl-pyrazine; and 2-ethyl-6-methyl-pyrazine could significantly increase the intensity of the caramel-like and roasted potato attributes; this is consistent with the results of mission analysis experiments [17,19]. Methional could significantly increase the intensity of the sauce-like and caramel-like aromas. Methional has been recently reported as a key aroma compound in Chinese soy sauces [19]. 2-Methoxy-phenol could significantly increase the smoky aroma; this result is consistent with the results of the task analysis experiments of Gao et al. [17]. Alcohol could significantly increase the intensity of alcoholic notes; this result is consistent with the results of the task analysis experiments of Gao et al. [17]. 2,3-Butanediol could increase the malty and alcoholic notes.

## 4. Conclusions

In this work, seven soy sauce samples that are suitable for cold dishes showed the highest preference scores, which were screened from six categories of soy sauce. A total of 38 aroma-active compounds were detected in these seven soy sauces via SPME-GC-MS/O. Among them, 2-methoxy-phenol (6–93); ethyl acetate (2–48); 3-methyl-1-butanol (4–30); 3-methyl-butanal (5–24); methional (0–22); dimethyl trisulfide (5–19); and dimethyl disulfide (0–8) showed higher ROAVs. Based on the PLSR and ROAV analysis, methional; 2-methoxy-phenol; ethanol; 2,5-dimethyl-pyrazine; and ethanol were characterized as the key aroma compounds in the soy sauces, contributing to the roasted potato, smoky, and caramel-like attributes. Additional tests further confirmed that the significant contributors to the roasted characteristic were 2,5-dimethyl-pyrazine; 2,6-dimethyl-pyrazine; and 2-ethyl-6-methyl-pyrazine. Methional had a significant contribution to the sauce-like note. Ethanol was a significant contributor to the alcoholic note. 2-Methoxy-phenol significantly contributed to the smoky note. 2,5-Dimethyl-pyrazine; methional; 2,6-dimethyl-pyrazine; and 2-ethyl-6-methyl-pyrazine significantly contributed to the caramel-like attribute. This study established an effective method for the rapid screening of applicable products with practical application scenarios, which help to improve the quality of soy sauce. In the future, the effects of soy sauce on the flavor improvements of different processing dishes (stewing, firing, cooking, etc.) should be further explored.

## Figures and Tables

**Figure 1 foods-12-03693-f001:**
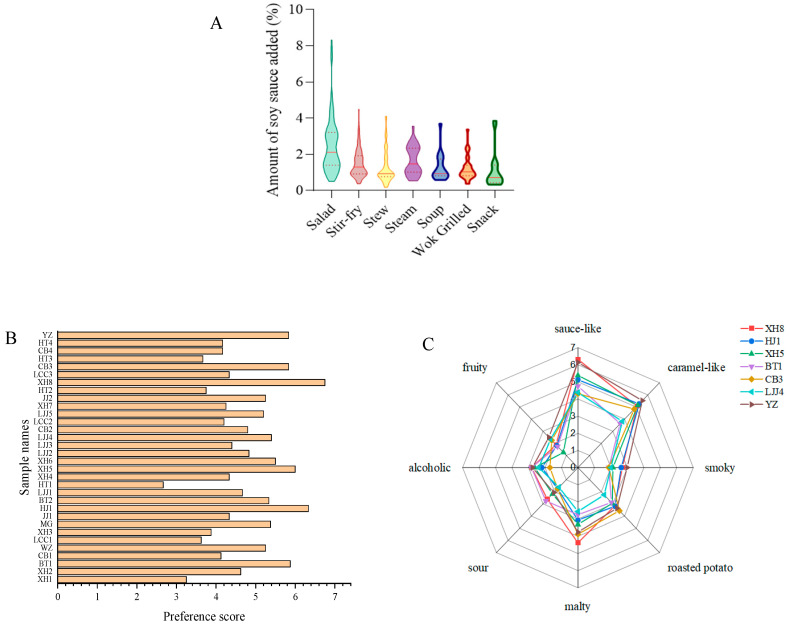
Sensory and statistical analysis results ((**A**), addition amount results of soy sauce in different kinds of dishes; (**B**), average ranking sum results of consumers’ preference of 32 soy sauces; (**C**), quantitative descriptive analysis results of seven kinds of soy sauce samples).

**Figure 2 foods-12-03693-f002:**
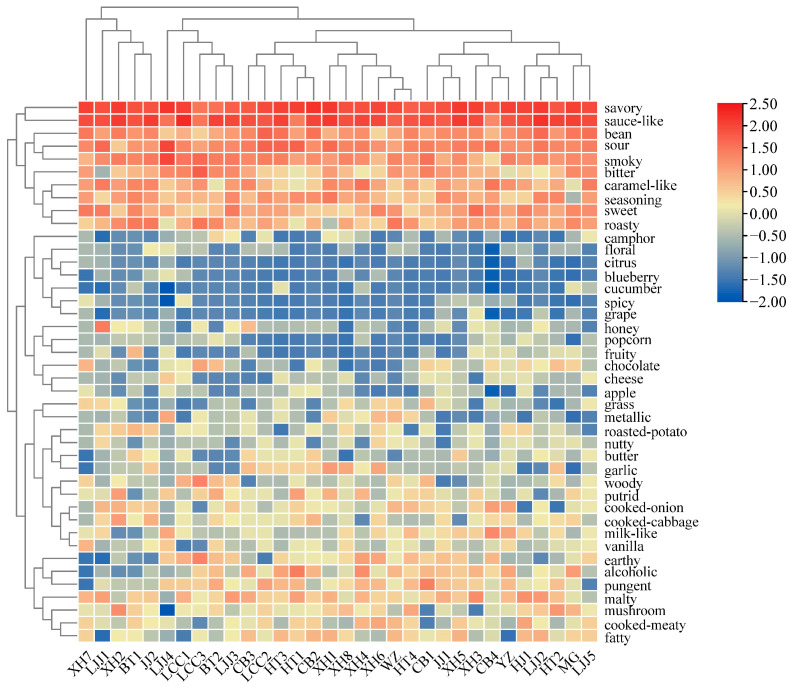
Heatmap of different brands of soy sauce samples by CATA analysis (The abbreviations of 32 kinds of soy sauce were showed in Table 1).

**Figure 3 foods-12-03693-f003:**
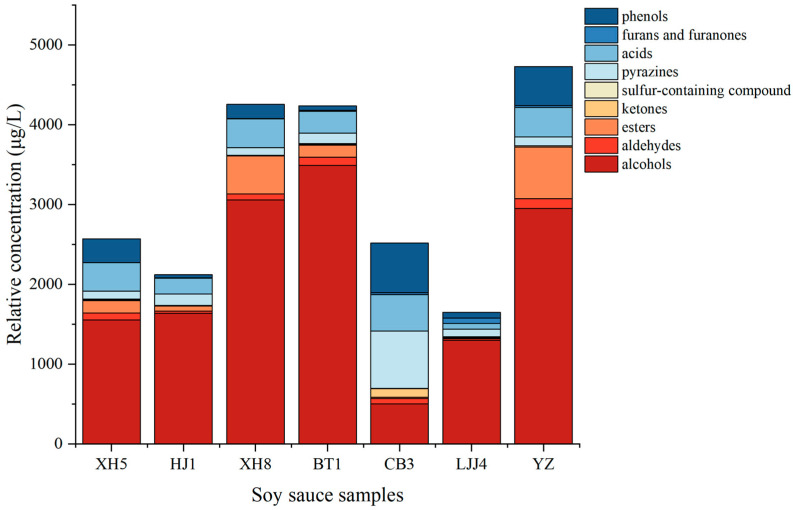
The relative concentration of the different types of aroma compounds in 7 different kinds of soy sauce.

**Figure 4 foods-12-03693-f004:**
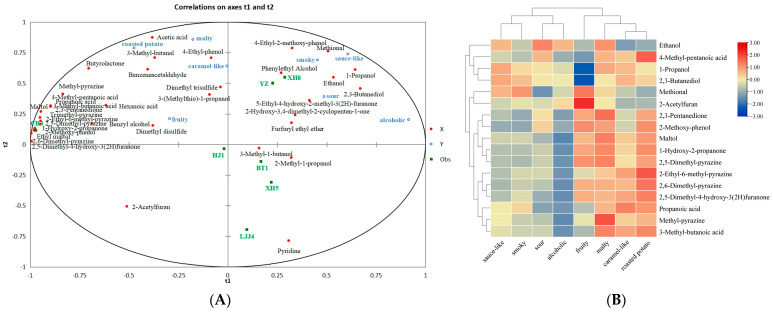
Results of PLSR analysis of aroma compounds and aroma attributes of seven soy sauces. ((**A**): The correlation matrix of the aroma-active compounds to sensory attributes. The red plots represent the 37 aroma-active compounds. The green plots represent the 7 soy sauce samples. The blue circles represent the 8 aroma attributes. (**B**): Heat map of the standard correlation coefficients of the aroma-active compounds to the aroma profiles.).

**Table 1 foods-12-03693-t001:** Identification and quantification of the aroma-active compounds in different soy sauce samples by SPME-GC-MS/O.

No.	Compounds	Aroma Quality	Identification Methods	RI	Relative Concentration (μg/L)						
					XH8	HJ	XH5	BT1	CB3	LJJ4	YZ
1	3-Methyl-butanal	Chocolate, malty	MS/RI/O	938/916	25.115 ± 0.442	20.463 ± 10.286	12.194 ± 0.905	24.323 ± 7.493	26.279 ± 2.886	5.353 ± 2.774	17.826 ± 3.435
2	Ethanol	Alcoholic	MS/RI/O	920/900	3854.470 ± 852.962	764.357 ± 118.160	892.693 ± 235.851	1790.055 ± 146.826	259.064 ± 38.294	734.591 ± 399.964	1273.804 ± 90.240
3	1-Propanol	Fermented, fusel, musty	MS/RI/O	999/1002	111.70 ± 37.02	38.95 ± 9.15	67.92 ± 12.060	32.29 ± 2.700	-	5.43 ± 1.040	63.55 ± 12.140
4	2,3-Pentanedione	Sweet, buttery, creamy	MS/RI/O	1070/1060	1.569 ± 1.029	0.691 ± 0.259	0.940 ± 0.798	1.484 ± 0.443	9.012 ± 0.527	0.765 ± 0.201	1.273 ± 0.150
5	Dimethyl disulfide	Sulfur compounds, cabbage, onion	MS/RI/O	1083/1078	2.537 ± 0.017	1.853 ± 0.650	-	2.470 ± 0.935	2.408 ± 0.789	1.182 ± 0.424	-
6	2-Methyl-1-propanol	Ethereal, wine, cortex	MS/RI/O	1071/1092	8.23 ± 2.15	-	22.71 ± 4.020	-	-	-	
7	Pyridine	Nauseating, fishy, amines	MS/RI/O	1141/1176	-	0.475 ± 0.105	0.590 ± 0.518	0.276 ± 0.025	-	0.417 ± 0.147	-
8	3-Methyl-1-butanol	Fusel, fruit, banana	MS/RI/O	1170/1185	50.488 ± 4.467	17.522 ± 2.437	20.602 ± 0.999	119.544 ± 22.598	40.536 ± 3.360	50.031 ± 1.642	45.330 ± 18.340
9	Methyl-pyrazine	Nuts, roasted, peanuts	MS/RI/O	1226/1261	15.271 ± 7.760	11.150 ± 2.976	2.675 ± 0.601	4.902 ± 1.739	27.676 ± 2.269	3.561 ± 0.173	4.952 ± 1.498
10	Furfuryl ethyl ether	Sweet, spicy	MS/RI/O	1247/1272	22.176 ± 1.793	9.570 ± 3.410	11.110 ± 2.809	42.907 ± 1.138	6.531 ± 0.227	2.309 ± 0.880	9.346 ± 3.771
11	1-Hydroxy-2-propanone	Sweet, caramel-like	MS/RI/O	1262/1275	1.431 ± 0.228	2.995 ± 0.436	1.463 ± 0.202	1.731 ± 0.641	40.644 ± 6.979	1.469 ± 0.271	1.477 ± 0.389
12	2,5-Dimethyl-pyrazine	Roasted, coffee	MS/RI/O	1290/1316	3.669 ± 1.256	4.259 ± 2.233	2.140 ± 0.379	2.356 ± 0.467	62.531 ± 4.202	3.770 ± 0.496	2.138 ± 0.708
13	2,6-Dimethyl-pyrazine	Roasted nuts, roast, coffee	MS/RI/O	1291/1319	9.884 ± 12.121	40.606 ± 8.141	9.412 ± 2.220	20.259 ± 3.903	165.814 ± 40.715	15.146 ± 0.924	12.440 ± 4.751
14	Dimethyl trisulfide	Sulfur, cooked onions	MS/RI/O	1334/1356	1.829 ± 0.018	1.902 ± 1.051	0.551 ± 0.133	1.578 ± 0.876	1.184 ± 0.079	0.502 ± 0.006	0.773 ± 0.168
15	2-Ethyl-6-methyl-pyrazine	Cooked potato	MS/RI/O	1350/1375	8.172 ± 1.430	22.623 ± 3.713	6.342 ± 0.200	13.234 ± 3.617	58.768 ± 14.615	5.470 ± 0.526	7.136 ± 1.828
16	Trimethyl-pyrazine	Roasted peanuts, hazelnuts	MS/RI/O	1370/1391	6.614 ± 0.498	13.703 ± 2.453	4.465 ± 0.380	9.405 ± 2.341	26.693 ± 6.591	4.301 ± 0.326	5.583 ± 1.414
17	Methional	Cooked-potato	MS/RI/O	1123/1145	8.045 ± 2.258	3.014 ± 0.888	-	3.444 ± 1.435	-	1.446 ± 0.591	9.877 ± 2.778
18	Acetic acid	Pungent and sour, vinegar	MS/RI/O	1414/1429	124.955 ± 13.541	84.402 ± 11.889	65.472 ± 19.132	64.866 ± 11.966	122.721 ± 30.649	19.807 ± 5.474	93.738 ± 4.253
19	2-Acetylfuran	Sweet, almond, coffee	MS/RI/O	1468/1483	-	-	-	3.324 ± 1.202	16.568 ± 1.557	22.862 ± 0.038	4.026 ± 2.241
20	Propanoic acid	Sour	MS/RI/O	1508/1498	2.384 ± 0.805	4.150 ± 1.531	1.226 ± 0.397	-	8.410 ± 2.126	0.296 ± 0.203	1.595 ± 0.117
21	2,3-Butanediol	Fruity, creamy, buttery	MS/RI/O	1555/1544	10.078 ± 5.300	8.895 ± 7.877	4.043 ± 0.956	6.032 ± 0.556	-	2.709 ± 1.236	5.802 ± 1.562
22	Butyrolactone	Creamy, caramel-like	MS/RI/O	1587/1602	2.392 ± 0.352	2.432 ± 2.333	1.082 ± 0.460	2.035 ± 0.543	3.783 ± 0.538	0.430 ± 0.608	1.949 ± 0.215
23	Benzeneacetaldehyde	Sweet, floral, honey	MS/RI/O	1599/1619	9.037 ± 1.737	4.391 ± 1.449	3.751 ± 0.867	16.591 ± 3.311	23.736 ± 8.520	1.676 ± 0.144	26.602 ± 3.600
24	3-Methyl-butanoic acid	Sweaty, cheese	MS/RI/O	1641/1647	9.148 ± 1.326	8.931 ± 2.442	6.815 ± 0.764	3.521 ± 0.862	39.130 ± 13.817	1.913 ± 0.527	4.094 ± 0.610
25	3-(Methylthio)-1-propanol	Sulfur, onions, vegetables	MS/RI/O	1687/1708	6.440 ± 0.058	10.189 ± 2.895	5.115 ± 1.896	9.399 ± 2.511	8.719 ± 4.038	7.206 ± 0.324	12.616 ± 1.448
26	4-Methyl-pentanoic acid	Spicy, cheese	MS/RI/O	1775/1800	3.368 ± 0.527	4.049 ± 1.092	1.435 ± 0.476	3.809 ± 1.084	10.219 ± 2.528	0.331 ± 0.061	1.919 ± 0.064
27	2-Hydroxy-3,4-dimethyl-2-cyclopenten-1-one	Caramel-like	MS/RI/O	1812/1839	-	1.070 ± 0.637	0.840 ± 0.073	3.603 ± 1.034	0.189 ± 0.327	0.233 ± 0.102	3.959 ± 1.375
28	Hexanoic acid	Sour, fatty, cheese	MS/RI/O	1817/1823	-	11.296 ± 9.634	-	2.919 ± 5.056	11.728 ± 3.319	1.384 ± 0.867	9.788 ± 2.301
29	2-Methoxy-phenol	Phenol, smoky, woody	MS/RI/O	1821/1823	3.341 ± 0.334	4.684 ± 0.338	2.759 ± 0.621	8.904 ± 2.692	44.862 ± 11.839	3.822 ± 0.372	4.604 ± 1.607
30	Benzyl alcohol	Floral, sweet	MS/RI/O	1847/1844	1.381 ± 0.980	2.933 ± 0.249	1.400 ± 0.356	5.342 ± 0.701	6.118 ± 1.455	1.375 ± 0.578	3.219 ± 0.650
31	Phenylethyl Alcohol	Floral, rose, sweet	MS/RI/O	1877/1875	120.135 ± 29.003	24.891 ± 1.248	31.737 ± 5.505	142.170 ± 32.563	63.832 ± 2.816	44.058 ± 2.041	125.174 ± 18.225
32	Maltol	Sweet, caramel-like, cotton candy	MS/RI/O	1927/1936	3.927 ± 0.628	3.162 ± 0.060	2.565 ± 0.980	2.683 ± 1.228	17.212 ± 4.162	1.970 ± 0.229	4.498 ± 1.791
33	Ethyl maltol	Sweet, caramel-like, cotton candy	MS/RI/O	1977/1989	-	-	-	-	221.431 ± 11.668	0.957 ± 0.157	0.572 ± 0.355
34	4-Ethyl-2-methoxy-phenol	Spicy, smoky	MS/RI/S/O	1989/1983	90.827 ± 26.452	12.986 ± 2.425	40.336 ± 7.857	5.648 ± 1.716	26.013 ± 7.275	10.171 ± 0.295	119.760 ± 12.688
35	2,5-Dimethyl-4-hydroxy-3(2H)furanone	Sweet, caramel-like, candy	MS/RI/S/O	2035/2039	0.074 ± 0.026	0.304 ± 0.181	0.216 ± 0.126	0.147 ± 0.085	1.176 ± 0.337	0.220 ± 0.020	0.189 ± 0.130
36	5-Ethyl-4-hydroxy-2-methyl-3(2H)-furanone	Sweet, caramel-like	MS/RI/S/O	2066/2090	1.308 ± 0.317	4.686 ± 3.186	4.823 ± 1.854	2.298 ± 0.863	0.415 ± 0.123	0.831 ± 0.029	9.776 ± 3.069
37	4-Ethyl-phenol	Phenol, smoky	MS/RI/O	2138/2153	13.335 ± 0.473	2.733 ± 0.689	3.152 ± 1.176	-	10.008 ± 5.550	6.216 ± 2.669	15.104 ± 3.855
38	Unknown compound	Chocolate		2152/-	-	-	-	-	-	-	-

“-” not detected; MS, identification by comparing the mass spectrometry (NIST 20); RI, identification by comparing the retention index; O, identification basecd on odor quality.

**Table 2 foods-12-03693-t002:** Results of the additional tests.

Added Compounds	Concentration (μg/L)	Change Trends in Intensity of Aroma Attributes							
		Sauce-Like	Caramel-Like	Smoky	Roasted Potato	Malty	Sour	Alcoholic	Fruity
2,5-Dimethyl-pyrazine (YZ)	60.392	↓	↑ *	-	↑ *	-	↓	↑	-
2,3-Butanediol (CB3)	10.078	-	-	-	-	↑	↓	↓	-
Methional (CB3)	9.877	↑ *	↑ *	-	↑	-	-	↑	-
Ethanol (CB3)	3595.406	↓	↓	-	↓	-	↓	↑ *	-
2,6-Dimethyl-pyrazine (XH8)	155.93	↓	↑ *	-	↑ *	-	↓	↑	-
2-Ethyl-6-methyl-pyrazine(LJJ4)	24.115	↓	↑ *	-	↑ *	-	↓	↑	-
2-Methoxy-phenol (XH5)	42.103	-	↓ *	↑ *	-	-	↓	↓	-

“↑” the aroma intensity increased; “↓” the aroma intensity decreased; *, significant difference at *p* < 0.05 (Duncan test).

## Data Availability

Data is contained within the article.

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
