# Peer review of "Decoding the Different Aroma-Active Compounds in Soy Sauce for Cold Dishes via a Multiple Sensory Evaluation and Instrumental Analysis"

_foods, 2023, doi:10.3390/foods12193693_

Round 1
Reviewer 1 Report (Previous Reviewer 3)
Although the manuscript has improved in the materials and methods section, the data presented still has gaps.
The authors wrote that they calculated the absolute concentrations of the compounds using an internal standard such as 2-methyl-3-heptanone.
Why did they choose this compound? Also, without determining a response factor for each compound, concentration values may not be correct. I highly recommend only determining a relative concentration expressed as a percentage.
Furthermore, with regard to the identification of the compounds, the authors calculated the Kovats index which for some compounds deviates by more than 10% from the value reported in the literature.
In this case the uncertainty in the identification is too high and therefore it is advisable to remove these compounds from the table.
Author Response
Reviewer 1
Dear editor and reviewers,
Thank you for your letter and for the reviewers’ comments concerning our manuscript. Those comments are very valuable and helpful for revising and improving our paper. We read all the comments carefully and made revisions which we hope meet with approval. The point-by-point responses to the reviewers’ comments are appended below. The detailed revisions with “Tracked Changes” are in the manuscript, and most of long sentences deleted or revised are marked red for better review.
Thank you for your patience once more. We are looking forward to your positive response.
Yours sincerely,
Dandan Pu
- Although the manuscript has improved in the materials and methods section, the data presented still has gaps.The authors wrote that they calculated the absolute concentrations of the compounds using an internal standard such as 2-methyl-3-heptanone.Why did they choose this compound? Also, without determining a response factor for each compound, concentration values may not be correct. I highly recommend only determining a relative concentration expressed as a percentage.
Response: Thank you for your suggestions. Firstly, the 2-methyl-3-heptanone was not existed in soy sauce sample. Secondly, there was no co-outflow of 2-methyl-3-heptanone with other compounds. Bedsides, the response signal was relatively good. Due to these advantages, this compound was widely used in other related articles:
(1) Wang X, Guo M, Song H, et al. Characterization of key odor-active compounds in commercial high-salt liquid-state soy sauce by switchable GC/GC× GC–olfactometry–MS and sensory evaluation[J]. Food Chemistry, 2021, 342: 128224.
(2) Wang X, Guo M, Song H, et al. Characterization of key aroma compounds in traditional Chinese soy sauce through the molecular sensory science technique[J]. Lwt, 2020, 128: 109413.
(3) Li J, Zhang M, Feng X, et al. Characterization of fragrant compounds in different types of high-salt liquid-state fermentation soy sauce from China[J]. LWT, 2022, 169: 113993.
The quantitative result was relative concentration, which has been changed to relative concentration in this work, and the aroma activity value has been changed to the relative odor activity value (ROAV).
- Furthermore, with regard to the identification of the compounds, the authors calculated the Kovats index which for some compounds deviates by more than 10% from the value reported in the literature.In this case the uncertainty in the identification is too high and therefore it is advisable to remove these compounds from the table.
Response: Thank you for your suggestions. In this work, we identified the aroma compounds based on the comparison of the calculated RI value to the referenced RI , and the difference between them is within 30. This difference value corresponds to a deviation of about 3%. The aroma compounds with higher variance values were checked carefully and revised in Table 2.

Reviewer 2 Report (New Reviewer)
foods-2571463
Decoding the soy sauce for cold dishes and the aroma-active compounds by Check-all-that-apply, quantitative description analysis and GC-MS/O analysis
Abstract: abstract should state briefly the purpose of the study undertaken, brief mention of experimental aspects (without using abbreviations), highlights of the results and important conclusions based on the obtained results. Therefore, it is suggested that the Abstract should be improved with more data.
L18 “XH8, HJ1, XH5, BT1, CB3, LJJ4, and YZ” → Abbreviations have to be written out at their first use.
Keywords→ arrange in alphabetical order.
Were total of 32 different kinds of soy sauce produced at same time?
The Materials and Methods and Results and Discussion section is somewhat confusingly structured and needs to be rewritten. The Authors first mention Figure 2 for the first time on pages 3, but they mention Figure 1 for the first time on pages 4. they continue to discuss this data in separate subsections through to page 6. Please separate the data in figures to follow a logical story, according to the subsections which you listed
Please introduce the sample codes in a table
Figures 1 and 3: Don’t you have better-quality figures?
Table 1 and Fig. 2: Where is other samples results?
In this paper, the methodology is well written, but discussion need to be improved. Results presented need a better discussion. There was no enough discussion or analysis of the results. There was no enough discussion or analysis of the results. The author should explain and clearly discuss this part based on scientific knowledge. It is better to compare the results with more similar recent works. And discuss the superiority of the work.
Author Response
Reviewer 2
Dear editor and reviewers,
Thank you for your letter and for the reviewers’ comments concerning our manuscript. Those comments are very valuable and helpful for revising and improving our paper. We read all the comments carefully and made revisions which we hope meet with approval. The point-by-point responses to the reviewers’ comments are appended below. The detailed revisions with “Tracked Changes” are in the manuscript, and most of long sentences deleted or revised are marked red for better review.
Thank you for your patience once more. We are looking forward to your positive response.
Yours sincerely,
Dandan Pu
- Decoding the soy sauce for cold dishes and the aroma-active compounds by Check-all-that-apply, quantitative description analysis and GC-MS/O analysis
Abstract: abstract should state briefly the purpose of the study undertaken, brief mention of experimental aspects (without using abbreviations), highlights of the results and important conclusions based on the obtained results. Therefore, it is suggested that the Abstract should be improved with more data.
Response: Thank you for your suggestions the abstract section was revised carefully.
Abstract: Multiple sensory evaluation and gas chromatography-mass spectrometry/olfactometry (GC-MS/O) analysis were combined to decode the soy sauces for cold dishes and characterize their differential aroma-active compounds. Thirty-two kinds of soy sauce with 42 sensory descriptors were classified into six categories by Check-all-that-apply and cluster analysis. The average soy sauce addition (1.74%) from 433 dishes in different cooking methods was summarized. Sensory evaluation results showed that seven soy sauce samples had the highest acceptance in each category. Solid-phase microextraction and solid phase extraction combined with GC-MS/O results showed that a total of 38 aroma-active compounds were identified, among which dimethyl trisulfide (5-19), 3-methyl-butanal (5-23), 5-ethyl-4-hydroxy-2-methyl-3(2H)-furanone (1-9), and 2-methoxy-phenol (6-93) showed higher relative odor activity value (ROAV). Partial least squares regression prediction combined with additional test further confirmed that 2,5-dimethyl-pyrazine, 2,6-dimethyl-pyrazine and 2-ethyl-6-methyl-pyrazine significantly contributed to roasted attribute, methional significantly contributed to sauce-like, ethanol significantly contributed to alcoholic note, 2-methoxy-phenol significantly contributed to smoky note. 2,5-Dimethyl-pyrazine, methional, 2,6-dimethyl-pyrazine and 2-ethyl-6-methyl-pyrazine significantly contributed to caramel-like.
- L18 “XH8, HJ1, XH5, BT1, CB3, LJJ4, and YZ” → Abbreviations have to be written out at their first use.
Response: Thank you for your suggestions. We have added the detail information abouth 32 soy sauce samples in the manuscript (chemical and materials section).
Table 1. Detail information of soy sauce samples.
|
Sample abbreviation |
Manufacturers (country) |
|
XH1 |
Yantai Shinho Enterprise Foods Co., Ltd. (China) |
|
XH2 |
|
|
XH3 |
|
|
XH4 |
|
|
XH5 |
|
|
XH6 |
|
|
XH7 |
|
|
XH8 |
|
|
YZ |
|
|
HT1 |
Foshan Haitian Flavouring and Food Co.,Ltd. (China) |
|
HT2 |
|
|
HT3 |
|
|
HT4 |
|
|
LJJ1 |
LEE KUM KEE (China) |
|
LJJ2 |
|
|
LJJ3 |
|
|
LJJ4 |
|
|
LJJ5 |
|
|
CB1 |
Guangdong Meiweixian Flavoring Foods Co.,Ltd. (China) |
|
CB2 |
|
|
CB3 |
|
|
CB4 |
|
|
JJ1 |
Jiajia Food Group Co.,Ltd.(China) |
|
JJ2 |
|
|
LCC1 |
Beijing Laocaichen Food Co.,Ltd. (China) |
|
LCC2 |
|
|
LCC3 |
|
|
BT1 |
Hamadaya (Japan) |
|
BT2 |
|
|
HJ |
Haoji Food Brewing Co., Ltd. (China) |
|
WZ |
KIKKOMAN (Japan) |
|
MG |
Nestle (Switzerland) |
- Keywords→ arrange in alphabetical order.
Response: Thank you for your suggestions. We have revised the keyword order in accordance with the modifications.
- Were total of 32 different kinds of soy sauce produced at same time?
Response: Thank you for your suggestions. The 32 samples were presented at the same time, but due to the relatively large number of samples, they were randomly divided into 4 groups and randomly numbered before presented to the panelists at the same time to avoid sequential interference.
- The Materials and Methods and Results and Discussion section is somewhat confusingly structured and needs to be rewritten. The Authors first mention Figure 2 for the first time on pages 3, but they mention Figure 1 for the first time on pages 4. they continue to discuss this data in separate subsections through to page 6. Please separate the data in figures to follow a logical story, according to the subsections which you listed. Please introduce the sample codes in a table
Response: Thank you for your suggestions. We have added information on 32 samples in the manuscript (Table 1).
Table 1. Detail information of soy sauce samples.
- Figures 1 and 3: Don’t you have better-quality figures?
Response: Thank you for your suggestions. We have replaced the images with higher resolution ones. The CATA analysis result were separately presented in Figure 1.
- Table 1 and Fig. 2: Where is other samples results?
Response: In the experiment, we first classified 32 soy sauces into 6 categories by CATA screening, and then selected the samples with the best consumer acceptance from each category, and a total of 7 samples were selected, and then focused on analyzing these 7 samples, and subjected the 7 samples to GC-MS/O analysis as well as PLSR analysis.
- In this paper, the methodology is well written, but discussion need to be improved. Results presented need a better discussion. There was no enough discussion or analysis of the results. There was no enough discussion or analysis of the results. The author should explain and clearly discuss this part based on scientific knowledge. It is better to compare the results with moresimilar recent works. And discuss the superiority of the work.
Response: Thank you for your suggestions, we have carefully revised and improved these sections in manuscript.
Line 397-409: “Addition tests were conducted to confirm the contribution of these aroma-active compounds to the aroma attributes. 7 odorants with high VIP values were selected and added into the sauce sample with the maximum relative concentration difference (Table 3). Addition tests results showed that 2,5-dimethyl-pyrazine, 2,6-dimethyl-pyrazine and 2-ethyl-6-methyl-pyrazine could significantly increase the intensity of caramel-like and roasted-potato attributes, this is consistent with the results of mission analysis experiments [17,19]. Methional could significantly increase the intensity of sauce-like and caramel-like aroma. The methional has been recently reported as a key aroma compound in Chinese soy sauces [19]. 2-Methoxy-phenol could significantly increase the smoky aroma, this result is consistent with the results of the task analysis experiments of Gao et al.[17]. Alcohol could significantly increase the intensity of alcoholic notes, this result is consistent with the results of the task analysis experiments of Gao et al.[17]. 2,3-Butanediol could increase malty and alcoholic.”

Reviewer 3 Report (New Reviewer)
This study aimed to detect the aroma profile variances among different brands of soy sauces.
Firstly, the manuscript is well organized, clearly states the aim, and the tables are well designed with information.
Other comments are reported below for the article:
1) My suggestion to the authors please give more detail about the categorization of 32 kinds of soy sauce samples were divided into 6 categories and analyzed 7 samples. This is the aroma profile study, why 32 samples were not analyzed?
2) I recommended authors to improve the quality of Figure 1A.
3) I recommend the authors improve the conclusion section, it is the same as the discussion section.
4) Why just 12 trained panelists?
5) While the relationship between sensory evaluation and the aroma compounds was discussed deeply. What is the innovation of this study from the published ones? Please explain in detail, with more references.
This study’s results may contribute to the development of novel methods for the development of the aroma profile of different sauces
Author Response
Reviewer 3
Dear editor and reviewers,
Thank you for your letter and for the reviewers’ comments concerning our manuscript. Those comments are very valuable and helpful for revising and improving our paper. We read all the comments carefully and made revisions which we hope meet with approval. The point-by-point responses to the reviewers’ comments are appended below. The detailed revisions with “Tracked Changes” are in the manuscript, and most of long sentences deleted or revised are marked red for better review.
Thank you for your patience once more. We are looking forward to your positive response.
Yours sincerely,
Dandan Pu
- This study aimed to detect the aroma profile variances among different brands of soy sauces.
Firstly, the manuscript is well organized, clearly states the aim, and the tables are well designed with information. This study’s results may contribute to the development of novel methods for the development of the aroma profile of different sauces
Response: Thank you for your affirmation. We improved the manuscript based on these useful comments.
- My suggestion to the authors please give more detail about the categorization of 32 kinds of soy sauce samples were divided into 6 categories and analyzed 7 samples. This is the aroma profile study, why 32 samples were not analyzed?
Response: Thank you for your suggestions. In this work, CATA analysis is performed to detect the aroma profile variances among different brands of soy sauce and to divide them into 6 different clusters. Then the application scenario of soy sauce in cold dishes is simulated (1.74%) to select the best acceptance samples from each cluster according to consumers’ preference (XH5, HJ1, XH8, BT1, CB3, YZ, and LJJ4). The aroma characteristics of the 7 favoritest soy sauce were carried out by quantitative descriptive analysis (QDA) to compare their differences. SPME-GC-MS and GC-MS/Olfactory (GC-MS/O) were combined to analyze the aroma-active compounds of the selected soy sauce. Finally, the partial least squares regulation analysis (PLSR) was performed to correlate the relationship between the aroma-active compounds to the sensory attributes and further clarify their contributions.
- I recommended authors to improve the quality of Figure 1A.
Response: Thank you for your suggestions. We have replaced the images with higher resolution ones and the CATA analysis results are analyzed separately.
- I recommend the authors improve the conclusion section, it is the same as the discussion section.
Response: Thank you for your suggestions. We have revised the manuscript in detail.
Line 418-437: “In this work, 6 categories sauce samples were divided by CATA analysis. The soy sauce was added into the pure water solution (1.74%) to simulate the cold dish preparation according to addition statistic of Chinese dishes. Seven soy sauce samples including XH8, HJ, XH5, BT2, CB3, LJJ3, and YZ soy sauce with the highest score of preference in each category were rated as soy sauce suitable for cold dish. A total of 38 aroma-active compounds were detected in these seven soy sauces by application of SPME-GC-MS/O. Dimethyl trisulfide (5-19), methional (0-22), dimethyl disulfide (0-8), ethyl acetate (2-48), 3-methyl-butanal (5-24), 3-methyl-1-butanol (4-30) and 2-methoxy-phenol (6-93) showed the higher ROAV. Based on the PLSR and ROAV analysis, methional, 2-methoxy-phenol, ethanol, 2,5-dimethyl-pyrazine and ethanol were characterized as the key aroma compounds in soy sauces suitable for the cold dish, contributing to the roasted-potato, smoky and caramel-like attributes. Addition tests further confirmed the Significant contributors to roasted flavor were 2,5-dimethyl-pyrazine, 2,6-dimethyl-pyrazine and 2-ethyl-6-methyl-pyrazine. Methional had significant contribution to sauce-like. Ethanol was Significant contributors to alcoholic note. 2-Methoxy-phenol significantly contributed to smoky note. 2,5-Dimethyl-pyrazine, methional, 2,6-dimethyl-pyrazine and 2-ethyl-6-methyl -pyrazineand significantly contributed to the caramel-like. This study established an effective method for rapid screening of applicable products, which help to select practical application scenarios and improve the quality of soy sauce in the future. In future, the effects of soy sauce on flavor improvement among different dishes in variance temperature-treated dishes (stewing, firing, cooking, etc.) should be further explored.”
- Why just 12 trained panelists?
Response: These 12 panelists were systematically trained in the laboratory and they were conducted weekly sensory evaluation tests such as aroma intensity, aroma profile, quantitative descriptive analysis, taste intensity perception resulting from the aroma induced, as well as dynamic sensory evaluation methods such as time intensity and temporal dominance of sensations.These panelists are highly experienced in identifying the aroma profiles and QDA analyses. Therefore, 12 trained panelists were used in this work.
- While the relationship between sensory evaluation and the aroma compounds was discussed deeply. What is the innovation of this study from the published ones? Please explain in detail, with more references.
Response: Firstly, most of the current papers performed the correlation analysis but lack of further validation test like follow references.
(1) Diez-Simon, C., Eichelsheim, C., Jacobs, D. M., Mumm, R., & Hall, R. D. Stir bar sorptive extraction of aroma compounds in soy sauce: Revealing the chemical diversity. Food Research International, 2021, 144, 963-969.
(2) Xiao Z, Fan B, Niu Y, et al. Characterization of odor-active compounds of various Chrysanthemum essential oils by gas chromatography–olfactometry, gas chromatography–mass spectrometry and their correlation with sensory attributes. Journal of Chromatography B, 2016, 1009: 152-162.
Secondly, this study compared with the published literature. Based on the content differences in actual soy sauce samples, the predicted results were further validated, confirming the contribution and importance of these differences in the components. These findings are providing an important guide for the quality improvement and adjustment of the actual product.

Reviewer 4 Report (New Reviewer)
In the paper " Decoding the soy sauce for cold dishes and the aroma-active compounds by Check-all-that-apply, quantitative description analysis and GC-MS/O analysis"
In my opinion it is a good study but need some modification the comments and questions are:
1-Unfortunately, the manuscript is dumb and irregular and needs to be written more clearly.
2-Abstract should be recognized according to journal categorizing.
3-You should clearly mention in the abstract or introduction the novelty of this article and the significance of approaching this topic.
4-The Introduction section is incomplete. Statistical methods, analytical methods as well as some comparison study should be added.
5-The chromatograms of 101 volatiles were detected by SPME-GC-MS should be added.
6-The large number of volatile analytes identified by GC-MS can cause peaks interference, so it is necessary to use standard for each analyte for accurate identification. You should use SIM mode instead of SCAN mode for all analytes.
7-Please give some information about the validation protocol to qualitative method (How confirmed the accuracy and precision). Since the standards of the analytes in the study have not been used to identify and determine, the method must be validated and the accuracy and precision of the method should be determined to ensure the acceptability of the results.
8-The reasons for choosing the type and concentration of the internal standard should be wrote.
9- The conclusion section is insufficient. The author should be explained in detail about the results.
Author Response
Reviewer 4
Dear editor and reviewers,
Thank you for your letter and for the reviewers’ comments concerning our manuscript. Those comments are very valuable and helpful for revising and improving our paper. We read all the comments carefully and made revisions which we hope meet with approval. The point-by-point responses to the reviewers’ comments are appended below. The detailed revisions with “Tracked Changes” are in the manuscript, and most of long sentences deleted or revised are marked red for better review.
Thank you for your patience once more. We are looking forward to your positive response.
Yours sincerely,
Dandan Pu
- In the paper " Decoding the soy sauce for cold dishes and the aroma-active compounds by Check-all-that-apply, quantitative description analysis and GC-MS/O analysis"In my opinion it is a good study but need some modification the comments and questions are:1-Unfortunately, the manuscript is dumb and irregular and needs to be written more clearly.
Response: Thank you for your suggestions, the abstract was revised.
Abstract: Multiple sensory evaluation and gas chromatography-mass spectrometry/olfactometry (GC-MS/O) analysis were combined to decode the soy sauces for cold dishes and characterize their differential aroma-active compounds. Thirty-two kinds of soy sauce with 42 sensory descriptors were classified into six categories by Check-all-that-apply and cluster analysis. The average soy sauce addition (1.74%) from 433 dishes in different cooking methods was summarized. Sensory evaluation results showed that seven soy sauce samples had the highest acceptance in each category. Solid-phase microextraction and solid phase extraction combined with GC-MS/O results showed that a total of 38 aroma-active compounds were identified, among which dimethyl trisulfide (5-19), 3-methyl-butanal (5-23), 5-ethyl-4-hydroxy-2-methyl-3(2H)-furanone (1-9), and 2-methoxy-phenol (6-93) showed higher relative odor activity value (ROAV). Partial least squares regression prediction combined with additional test further confirmed that 2,5-dimethyl-pyrazine, 2,6-dimethyl-pyrazine and 2-ethyl-6-methyl-pyrazine significantly contributed to roasted attribute, methional significantly contributed to sauce-like, ethanol significantly contributed to alcoholic note, 2-methoxy-phenol significantly contributed to smoky note. 2,5-Dimethyl-pyrazine, methional, 2,6-dimethyl-pyrazine and 2-ethyl-6-methyl-pyrazine significantly contributed to caramel-like..
Line 71-78: “Additionally, to correlate the relationship between the sensory property and chemical data, many multiple analysis methods were performed. For example, Li et al. [18] determined the variances among different types, processes, and regions of soy sauce by principal component analysis (PCA), hierarchical cluster analysis (HAC) and orthogonal partial least squares discriminant analysis (OPLS-DA). Diez-Simon et al. [19] comprehensively characterized the volatile profiles and related compositional differences in the volatile profiles to the origin and production history of the samples by PCA, Heatmap and partial least squares-discriminant analysis(PLS-DA). ”
Line 398-410: “Addition tests were conducted to confirm the contribution of these aroma-active compounds to the aroma attributes. 7 odorants with high VIP values were selected and added into the sauce sample with the maximum relative concentration difference (Table 3). Addition tests results showed that 2,5-dimethyl-pyrazine, 2,6-dimethyl-pyrazine and 2-ethyl-6-methyl-pyrazine could significantly increase the intensity of caramel-like and roasted-potato attributes, this is consistent with the results of mission analysis experiments [17,19]. Methional could significantly increase the intensity of sauce-like and caramel-like aroma. The methional has been recently reported as a key aroma compound in Chinese soy sauces [19]. 2-Methoxy-phenol could significantly increase the smoky aroma, this result is consistent with the results of the task analysis experiments of Gao et al.[17]. Alcohol could significantly increase the intensity of alcoholic notes, this result is consistent with the results of the task analysis experiments of Gao et al.[17]. 2,3-Butanediol could increase malty and alcoholic.”
Line 418-437: “In this work, 6 categories sauce samples were divided by CATA analysis. The soy sauce was added into the pure water solution (1.74%) to simulate the cold dish preparation according to addition statistic of Chinese dishes. Seven soy sauce samples including XH8, HJ, XH5, BT2, CB3, LJJ3, and YZ soy sauce with the highest score of preference in each category were rated as soy sauce suitable for cold dish. A total of 38 aroma-active compounds were detected in these seven soy sauces by application of SPME-GC-MS/O. Dimethyl trisulfide (5-19), methional (0-22), dimethyl disulfide (0-8), ethyl acetate (2-48), 3-methyl-butanal (5-24), 3-methyl-1-butanol (4-30) and 2-methoxy-phenol (6-93) showed the higher ROAV. Based on the PLSR and ROAV analysis, methional, 2-methoxy-phenol, ethanol, 2,5-dimethyl-pyrazine and ethanol were characterized as the key aroma compounds in soy sauces suitable for the cold dish, contributing to the roasted-potato, smoky and caramel-like attributes. Addition tests further confirmed the Significant contributors to roasted flavor were 2,5-dimethyl-pyrazine, 2,6-dimethyl-pyrazine and 2-ethyl-6-methyl-pyrazine. Methional had significant contribution to sauce-like. Ethanol was Significant contributors to alcoholic note. 2-Methoxy-phenol significantly contributed to smoky note. 2,5-Dimethyl-pyrazine, methional, 2,6-dimethyl-pyrazine and 2-ethyl-6-methyl -pyrazineand significantly contributed to the caramel-like. This study established an effective method for rapid screening of applicable products, which help to select practical application scenarios and improve the quality of soy sauce in the future. In future, the effects of soy sauce on flavor improvement among different dishes in variance temperature-treated dishes (stewing, firing, cooking, etc.) should be further explored.”
- 2. Abstract should be recognized according tojournal categorizing.
Response: Thank you for your suggestions. Type of the Paper has been added before the article title.
- 3. You should clearly mention in the abstract or introduction the novelty of this article and the significance of approaching this topic.
Response: Thank you for your suggestions. This work established an effective method for rapid screening of applicable products, which help to select practical application scenarios and improve the quality of soy sauce in the future. We have made changes in the Introduction section of the text (line 80-82).
- 4. The Introduction section is incomplete. Statistical methods, analytical methods as well as some comparison study should be added.
Response: Thank you for your suggestions. We have added the methodology of the statistical analysis in the body of the introduction kind based on the comments.
Line 73-80: “Additionally, to correlate the relationship between the sensory property and chemical data, many multiple analysis methods were performed. For example, Li et al. [18] determined the variances among different types, processes, and regions of soy sauce by principal component analysis (PCA), hierarchical cluster analysis (HAC) and orthogonal partial least squares discriminant analysis (OPLS-DA). Diez-Simon et al. [19] comprehensively characterized the volatile profiles and related compositional differences in the volatile profiles to the origin and production history of the samples by PCA, Heatmap and partial least squares-discriminant analysis(PLS-DA). ”
- 5. The chromatograms of 101 volatiles were detected by SPME-GC-MS should be added.
Response: Figure S1. Total ion chromatogram of volatile components in soy sauce samples analysis by GC-MS
Response: Thank you for your suggestions. Due to the large number of images in the main text, we have added the chromatograms in the supplementary material.
- 6. The large number of volatile analytes identified by GC-MS can cause peaks interference, so it is necessary to use standard for each analyte for accurate identification. You should use SIM mode instead of SCAN mode for all analytes.
Response: Thank you for your suggestions. The aim of our paper is to screen soy sauce from a large number of samples to be suitable for a certain cooking style, and then to find the differential components based on their relative contents, and to verify the contribution of important differential components to the sensory properties of soy sauce by addition experiments. For qualitative and quantitative analysis, we used semi-quantitative methods. Semi-quantitative allows access to both qualitative and quantitative data, allowing for a more complete understanding of the research subject.
(1) Wang Y, Li C, Zhao Y, et al. Novel insight into the formation mechanism of volatile flavor in Chinese fish sauce (Yu-lu) based on molecular sensory and metagenomics analyses[J]. Food Chemistry, 2020, 323: 126839.
(2) Zhao G, Ding L L, Hadiatullah H, et al. Characterization of the typical fragrant compounds in traditional Chinese-type soy sauce[J]. Food chemistry, 2020, 312: 126054.
(3) Zhang L, Chen J, Zhang J, et al. Lipid oxidation in fragrant rapeseed oil: Impact of seed roasting on the generation of key volatile compounds[J]. Food Chemistry: X, 2022, 16: 100491.
We have modified the content and OAV to relative content and relative ROAV values. This method has also been successful applied in helping us to quickly and efficiently find out the key differential compounds for flavor differences in different soy sauces.
We used the standard characterization and SIM methods mentioned by the reviewer in subsequent sample analysis experiments.
- 7. Please give some information about the validation protocol to qualitative method (How confirmed the accuracy and precision). Since the standards of the analytes in the study have not been used to identify and determine, the method must be validated and the accuracy and precision of the method should be determined to ensure the acceptability of the results.
Response: Thank you for your suggestions. A column for identification methods was added in Table 2: MS/RI/S/O.The internal standard deviation is within 10% (peak area), which validates the accuracy of the method.
- 8. The reasons for choosing the type and concentration of the internal standard should be wrote.
Response: Thank you for your suggestions. The soy sauce sample did not internalize 2-methyl-3-heptanone, there was no overlap/masking with other compounds, and the response signal was relatively good. This was verified in other related articles with the following references.
- Wang X, Guo M, Song H, et al. Characterization of key odor-active compounds in commercial high-salt liquid-state soy sauce by switchable GC/GC× GC–olfactometry–MS and sensory evaluation[J]. Food Chemistry, 2021, 342: 128224.
- Wang X, Guo M, Song H, et al. Characterization of key aroma compounds in traditional Chinese soy sauce through the molecular sensory science technique[J]. Lwt, 2020, 128: 109413.
- Li J, Zhang M, Feng X, et al. Characterization of fragrant compounds in different types of high-salt liquid-state fermentation soy sauce from China[J]. LWT, 2022, 169: 113993.
- 9. The conclusion section is insufficient. The author should be explained in detail about the results.
Response: Thank you for your suggestions. We have revised the conclusion section in detail.
Line418-437: “In this work, 6 categories sauce samples were divided by CATA analysis. The soy sauce was added into the pure water solution (1.74%) to simulate the cold dish preparation according to addition statistic of Chinese dishes. Seven soy sauce samples including XH8, HJ, XH5, BT2, CB3, LJJ3, and YZ soy sauce with the highest score of preference in each category were rated as soy sauce suitable for cold dish. A total of 38 aroma-active compounds were detected in these seven soy sauces by application of SPME-GC-MS/O. Dimethyl trisulfide (5-19), methional (0-22), dimethyl disulfide (0-8), ethyl acetate (2-48), 3-methyl-butanal (5-24), 3-methyl-1-butanol (4-30) and 2-methoxy-phenol (6-93) showed the higher ROAV. Based on the PLSR and ROAV analysis, methional, 2-methoxy-phenol, ethanol, 2,5-dimethyl-pyrazine and ethanol were characterized as the key aroma compounds in soy sauces suitable for the cold dish, contributing to the roasted-potato, smoky and caramel-like attributes. Addition tests further confirmed the Significant contributors to roasted flavor were 2,5-dimethyl-pyrazine, 2,6-dimethyl-pyrazine and 2-ethyl-6-methyl-pyrazine. Methional had significant contribution to sauce-like. Ethanol was Significant contributors to alcoholic note. 2-Methoxy-phenol significantly contributed to smoky note. 2,5-Dimethyl-pyrazine, methional, 2,6-dimethyl-pyrazine and 2-ethyl-6-methyl -pyrazineand significantly contributed to the caramel-like. This study established an effective method for rapid screening of applicable products, which help to select practical application scenarios and improve the quality of soy sauce in the future. In future, the effects of soy sauce on flavor improvement among different dishes in variance temperature-treated dishes (stewing, firing, cooking, etc.) should be further explored.”

Round 2
Reviewer 1 Report (Previous Reviewer 3)
The authors have answered the final questions and adjusted the retention rates as required.
The work can now be accepted for publication
Author Response
Thanks for your comments, we have carefully revised the manuscript and improved the quality of English expression.

Reviewer 2 Report (New Reviewer)
No comment
Author Response
Thanks for your comments, we have checked the whole manuscript and improved the expression quality.

Reviewer 4 Report (New Reviewer)
Dear editor
The requested modified point are approximately done but the abstract is not well categorized, without introduction, conclusion is long .....
Author Response
Dear editor and reviewers,
Thank you for your letter and for the reviewers’ comments concerning our manuscript. Those comments are very valuable and helpful for revising and improving our paper. We read all the comments carefully and made revisions which we hope meet with approval. The point-by-point responses to the reviewers’ comments are appended below. The detailed revisions with “Tracked Changes” are in the manuscript, and most of long sentences deleted or revised are marked red for better review.
Thank you for your patience once more. We are looking forward to your positive response.
Yours sincerely,
Dandan Pu
1. The requested modified point are approximately done but the abstract is not well categorized, without introduction, conclusion is long .....
Response: The abstract was revised and the back ground introduction was added. The conclusion section was shorten within 190 words.
Abstract: Screening soy sauce suitable for specific cooking method from variance products is beneficial for the fine development of soy sauce industry. Multiple sensory evaluation and gas chromatography-mass spectrometry/olfactometry (GC-MS/O) analysis were combined to decode the soy sauces for cold dishes and characterize their differential aroma-active compounds. Thirty-two kinds of soy sauce with 42 sensory descriptors were determined by Check-all-that-apply analysis and further classified into six categories by cluster analysis. Sensory evaluation results showed that seven soy sauce samples had the highest acceptance in each category. Solid-phase microextraction and solid phase extraction combined with GC-MS/O analysis results showed that a total of 38 aroma-active compounds were identified in seven soy sauce samples, among which 2-methoxy-phenol (6-93), ethyl acetate (2-48), 3-methyl-1-butanol (4-30), 3-methyl-butanal (5-24), methional (0-22), dimethyl trisulfide (5-19) and dimethyl disulfide (0-8) showed higher relative odor activity value (ROAV). Partial least squares regression prediction combined with additional test further confirmed that 2,5-dimethyl-pyrazine, 2,6-dimethyl-pyrazine and 2-ethyl-6-methyl-pyrazine significantly contributed to roasted attribute, methional significantly contributed to sauce-like note, ethanol significantly contributed to alcoholic note, 2-methoxy-phenol significantly contributed to smoky note. 2,5-Dimethyl-pyrazine, methional, 2,6-dimethyl-pyrazine and 2-ethyl-6-methyl-pyrazine significantly contributed to caramel-like attribute.
Conclusion: In this work, seven soy sauce samples suitable for cold dish showed the highest preference score were screened from 6 categories of soy sauce. A total of 38 aroma-active compounds were detected in these seven soy sauces by SPME-GC-MS/O. Among them, 2-methoxy-phenol (6-93), ethyl acetate (2-48), 3-methyl-1-butanol (4-30), 3-methyl-butanal (5-24), methional (0-22), dimethyl trisulfide (5-19) and dimethyl disulfide (0-8) showed the higher ROAV. Based on the PLSR and ROAV analysis, methional, 2-methoxy-phenol, ethanol, 2,5-dimethyl-pyrazine and ethanol were characterized as the key aroma compounds in soy sauces, contributing to the roasted-potato, smoky, and caramel-like attributes. Addition tests further confirmed the significant contributors to roasted characteristic were 2,5-dimethyl-pyrazine, 2,6-dimethyl-pyrazine and 2-ethyl-6-methyl-pyrazine. Methional had significant contribution to sauce-like note. Ethanol was Significant contributors to alcoholic note. 2-Methoxy-phenol significantly contributed to smoky note. 2,5-Dimethyl-pyrazine, methional, 2,6-dimethyl-pyrazine and 2-ethyl-6-methyl-pyrazine significantly contributed to the caramel-like attribute. This study established an effective method for rapid screening of applicable products with practical application scenarios which help to improve the quality of soy sauce. In future, the effects of soy sauce on flavor improvement of different processing dishes (stewing, firing, cooking, etc.) should be further explored.

This manuscript is a resubmission of an earlier submission. The following is a list of the peer review reports and author responses from that submission.
Round 1
Reviewer 1 Report
The manuscript entitled: "Decoding the soy sauce for cold dishes and the aroma-active compounds by Check-all-that-apply, quantitative description analysis and GC-MS/O analysis" presented the results of sensory analysis and chromatographic research of soy sauce in the matrix.
Few comments: the keywords should differ from those used in the title. I also recommend not using CAS numbers in the abstract, but paying more attention to the presentation of results. Moreover, there is no significant description of the aim and what was achieved during this study. "will help to select practical application scenarios and improve the quality of soy sauce in the future" It is not enough to outcomes of this research.
The methodology described in subchapter 2.2 "Sensory evaluation" should be checked once more. It is more likely that the first description is done for QDA and the second for CATA. Due to my knowledge, CATA is dedicated to consumer analysis, which means judges without previous training, whereas the QDA test is done mostly by trained experts. Figure 2A should be presented just below the text that described its content.
If the calculation for volatile compounds was done only according to internal standards, it cannot be named "concentration" but "relative content". What was the reason for using 2-methyl-3-heptanone as a standard? The list of standards seems not to be fulfilled.
I advise firstly to precisely present and discuss the GC-MS/O analysis and then QDA and CATA profiles. However, I have doubts about the proper CATA methodology. For sure QDA doesn't reflect consumer study.
Author Response
Dear editor and reviewers,
Thank you for your letter and for the reviewers’ comments concerning our manuscript. Those comments are very valuable and helpful for revising and improving our paper. We read all the comments carefully and made revisions which we hope meet with approval. The point-by-point responses to the reviewers’ comments are appended below. The detailed revisions with “Tracked Changes” are in the manuscript, and most of long sentences deleted or revised are marked red for better review.
Thank you for your patience once more. We are looking forward to your positive response.
Yours sincerely,
Dandan Pu
Reviewer: 1
Comments:
The manuscript entitled, “Decoding the soy sauce for cold dishes and the aroma-active compounds by Check-all-that-apply, quantitative description analysis and GC-MS/O analysis” examine the aroma profile differences among various brands of soy sauce. Furthermore, the authors have performed the partial least squares regulation analysis (PLSR) to correlate the relationship between the aroma-active compounds to the sensory attributes. However, some major concerns should be addressed by the authors.
Response: Thank you for your useful suggestion, it’s important for improving the quality of our manuscript. We have carefully revised our manuscripts according to the suggestions.
- The ‘title’ of the manuscript is vague and confusing and the authors may think about any short and crisp title for this article. The abbreviated words should be avoided in the title.
Response: the title was revised to “Rapid screening protocol of the soy sauce for cold dishes and identification of the aroma-active compounds”
- In the ‘Materials and Methods’ section, the authors have mentioned, “A total of 31 different kinds of soy sauce…...” However, in the ‘Abstract’ and ‘Results’ section, 32 kinds of soy sauce samples are mentioned. Which one correct?
Response: Thank you. The soy sauce sample is 32, including 31 purchased from the supermarket and one provided by Yantai Shinho Food Co., Ltd. We have revised the 31 to 32 at the Materials and Chemicals.
- The #Figure_1A is of very poor quality and thus is unclear. It should be corrected.
Response: Thank you. The Figure 1A of the CATA analysis was revised to Figure 1 and the quality was improved.
Figure 1. Heatmap of different brands of soy sauce samples by CATA analysis (the abbreviation of horizontal ordinate represented the name of 32 kinds of soy sauce presented in 2.1 Materials and Chemicals section).
- On what basis the authors have chosen only 7 soy sauce samples for SPME-GC-MS/O analysis and the data presented in the #Figure_2?
Response: Seven soy sauce samples with the highest consumers’ preference were selected for SPME-GC-MS/O analysis. Firstly, the Check-all-that-apply (CATA) analysis was conducted to evaluation the sensory quality of 32 soy sauce samples. Then, the cluster analysis was performed to classify the different cluster of 32 soy sauce samples and 6 categories were obtained (Figure 1). Category 1: HJ, LJJ2, HT2, MG and LJJ5 soy sauce; Category 2: CB1, JJ1, XH5, XH3, CB4 and YZ soy sauce; Category 3: XH1, XH8, WZ, HT4, XH4 and XH6 soy sauce; Category 4: CB3, LCC2, HT3, HT1 and CB2 Soy Sauce; Category 5: LJJ4, LCC1, LCC3, BT2 and LJJ3 soy sauce; Category 6: LJJ1, XH2, BT1 and JJ2 soy sauce. Subsequently, each sample in different cluster was added into pure water (1.74%, Figure 2A) in simulation of cold dishes, and further evaluated by consumers’ preference (Figure 2B). Finally, 7 soy sauce samples (Figure 2C) were selected for QDA analysis and SPME-GC-MS/O analysis.
- All the tests performed (data presented in #Table_2) should be clarified in the ‘Materials and Methods’ section and footnote of #Table_2.
Response: the sentence of sample abbreviation was added in Materials and Methods’ section: “The abbreviations of the soy sample were: WZ, MG, HJ, CB1, CB2, CB3, CB4, XH1, XH2, XH3, XH4, XH5, XH6, XH7, XH8, HT1, HT2, HT3, HT4, LCC1, LCC2, LCC3, LJJ1, LJJ2, LJJ3, LJJ4, LJJ5, BT1, BT2, JJ1, JJ2 and YZ.”.
Table 1: The names of 7 soy sauce samples presented in Table 1 and RI values were also attached at the footnote: “a, calculated retention index (RI); b, RI reference from the literature; the abbreviation of XH5, XH8, HJ, BT1, CB3, LJJ4, and YZ represented 7 kinds of soy sauce”
Table 2: The footnote of Table 2 was added: “the abbreviation of BT, CB3, XH5, YZ, HJ, LJJ4 and XH8 represented 7 kinds of soy sauce”.
- The abbreviated words in the legends (Figures and Tables) should be explained.
Response: The Figure and Table legends were all revised as follows:
Figure 1. Heatmap of different brands of soy sauce samples by CATA analysis (the abbreviation of horizontal ordinate represented the name of 32 kinds of soy sauce presented in 2.1 Materials and Chemicals section).
Figure 2. Statistical and sensory evaluation results (A, Addition amount results of soy sauce in different dishes; B, Average ranking sum results of consumers' preference of the 32 kinds of soy sauce, the abbreviation names loaded in Y-axis represented the 32 kinds of soy sauce; C, Quantitative descriptive analysis results of 7 kinds of soy sauce sample).
Figure 3. The content of different types of the aroma compounds in 7 different types of soy sauce (the horizontal ordinate was the abbreviation name of 32 kinds of soy sauce).
Figure 4. Results of PLSR analysis of aroma compounds and aroma attributes of seven soy sauces. (A. The correlation matrix of the aroma-active compounds to sensory attributes. The red plots represent the 37 aroma-active compounds with OAV ≥ 1. The green plots represent the 7 soy sauce samples. The blue circles represent the 8 aroma attributes. B. Heat map of standard correlation coefficients of the aroma-active compounds to the aroma profiles. The abbreviation of XH5, XH8, HJ, BT1, CB3, LJJ4, and YZ represented 7 kinds of soy sauce.).
- What does mean by the statement written in #line_391-392 (“A total of 37 aroma-active compounds were detected in these seven soy sauces by application of SPME-GC-MS/O, 27 of which were both detected in seven samples”) of ‘Conclusion’ section?
Response: The sentence was revised to “A total of 37 aroma-active compounds were detected in soy sauces by SPME-GC-MS/O, and 27 of them were detected in all samples”.
- The practical application and limitations of this study should be highlighted in the ‘Conclusion’ section.
Response: practical application and limitations of this work were added:“In the future, the formation or derivation mechanisms of these representative aroma com-pounds during soy sauce fermentation or shelf life storage should be further investigated. ”.
Reviewer 2 Report
The manuscript entitled, “Decoding the soy sauce for cold dishes and the aroma-active compounds by Check-all-that-apply, quantitative description analysis and GC-MS/O analysis” examine the aroma profile differences among various brands of soy sauce. Furthermore, the authors have performed the partial least squares regulation analysis (PLSR) to correlate the relationship between the aroma-active compounds to the sensory attributes. However, some major concerns should be addressed by the authors.
1. The ‘title’ of the manuscript is vague and confusing and the authors may think about any short and crisp title for this article. The abbreviated words should be avoided in the title.
2. In the ‘Materials and Methods’ section, the authors have mentioned, “A total of 31 different kinds of soy sauce…...” However, in the ‘Abstract’ and ‘Results’ section, 32 kinds of soy sauce samples are mentioned. Which one correct?
3. The #Figure_1A is of very poor quality and thus is unclear. It should be corrected.
4. On what basis the authors have chosen only 7 soy sauce samples for SPME-GC-MS/O analysis and the data presented in the #Figure_2?
5. All the tests performed (data presented in #Table_2) should be clarified in the ‘Materials and Methods’ section and footnote of #Table_2.
6. The abbreviated words in the legends (Figures and Tables) should be explained.
7. What does mean by the statement written in #line_391-392 (“A total of 37 aroma-active compounds were detected in these seven soy sauces by application of SPME-GC-MS/O, 27 of which were both detected in seven samples”) of ‘Conclusion’ section?
8. The practical application and limitations of this study should be highlighted in the ‘Conclusion’ section.
Moderate editing of English language
Author Response
Dear editor and reviewers,
Thank you for your letter and for the reviewers’ comments concerning our manuscript. Those comments are very valuable and helpful for revising and improving our paper. We read all the comments carefully and made revisions which we hope meet with approval. The point-by-point responses to the reviewers’ comments are appended below. The detailed revisions with “Tracked Changes” are in the manuscript, and most of long sentences deleted or revised are marked red for better review.
Thank you for your patience once more. We are looking forward to your positive response.
Yours sincerely,
Dandan Pu
Reviewer: 2
The manuscript by Dandan Pu and colleagues is very interesting. Apply the GC_MS_O for flavor profile characterization of soy sauces. The applied experimental method is well written and detailed. The results are clear and can be better interpreted thanks to an accurate statistical analysis. I advise the authors to insert two separate columns in table 1 to indicate the RIs deriving from the literature and those calculated as this is not clear in the way they are currently presented.
Response: Thank you for your useful suggestion, we have divided the RI into two lists (RIa and RIb), and the footnote of Table 1 were added: “a, calculated retention index (RI); b, RI reference from the literature”.

Reviewer 3 Report
The manuscript by Dandan Pu and colleagues is very interesting.
Apply the GC_MS_O for flavor profile characterization of soy sauces.
The applied experimental method is well written and detailed.
The results are clear and can be better interpreted thanks to an accurate statistical analysis.
I advise the authors to insert two separate columns in table 1 to indicate the RIs deriving from the literature and those calculated as this is not clear in the way they are currently presented.
Author Response
Dear editor and reviewers,
Thank you for your letter and for the reviewers’ comments concerning our manuscript. Those comments are very valuable and helpful for revising and improving our paper. We read all the comments carefully and made revisions which we hope meet with approval. The point-by-point responses to the reviewers’ comments are appended below. The detailed revisions with “Tracked Changes” are in the manuscript, and most of long sentences deleted or revised are marked red for better review.
Thank you for your patience once more. We are looking forward to your positive response.
Yours sincerely,
Dandan Pu
Reviewer: 3
The Manuscript describes new and relevant information about the aromatic compounds of soy sauces. The experimental assays have been correctly designed, performed, described and interpreted. The multivariate statistical analysis has been well developed and depicted in Figures and Tables of the Manuscript. The conclusions are supported by the experimental results.
Response: Thanks for your comments. We have carefully revised the expression, footnote of Table and Figure legend to improve the quality of manuscript.

Reviewer 4 Report
The Manuscript describes new and relevant information about the aromatic compounds of soy sauces.
The experimental assays have been correctly designed, performed, described and interpreted. The multivariate statistical analysis has been well developed and depicted in Figures and Tables of the Manuscript. The conclusions are supported by the experimental results.
Best regards.
Author Response
Dear editor and reviewers,
Thank you for your letter and for the reviewers’ comments concerning our manuscript. Those comments are very valuable and helpful for revising and improving our paper. We read all the comments carefully and made revisions which we hope meet with approval. The point-by-point responses to the reviewers’ comments are appended below. The detailed revisions with “Tracked Changes” are in the manuscript, and most of long sentences deleted or revised are marked red for better review.
Thank you for your patience once more. We are looking forward to your positive response.
Yours sincerely,
Dandan Pu
Reviewer: 4
The Manuscript describes new and relevant information about the aromatic compounds of soy sauces.The experimental assays have been correctly designed, performed, described and interpreted. The multivariate statistical analysis has been well developed and depicted in Figures and Tables of the Manuscript. The conclusions are supported by the experimental results.
Response: Thanks for your comments. We have carefully revised the expression, footnote of Table and Figure legend to improve the quality of manuscript.

Round 2
Reviewer 1 Report
I didn't find answers to my comments in the reviewed manuscript and in the answers.
Author Response
Dear editor and reviewers,
Thank you for your letter and for the reviewers’ comments concerning our manuscript. Those comments are very valuable and helpful for revising and improving our paper. We read all the comments carefully and made revisions which we hope meet with approval. The point-by-point responses to the reviewers’ comments are appended below. The detailed revisions with “Tracked Changes” are in the manuscript, and most of long sentences deleted or revised are marked red for better review.
Thank you for your patience once more. We are looking forward to your positive response.
Yours sincerely,
Dandan Pu
Reviewer: 1
- "The manuscript entitled: "Decoding the soy sauce for cold dishes and the aroma-active compounds by Check-all-that-apply, quantitative description analysis and GC-MS/O analysis" presented the results of sensory analysis and chromatographic research of soy sauce in the matrix.
Few comments: the keywords should differ from those used in the title. I also recommend not using CAS numbers in the abstract, but paying more attention to the presentation of results. Moreover, there is no significant description of the aim and what was achieved during this study. "will help to select practical application scenarios and improve the quality of soy sauce in the future" It is not enough to outcomes of this research.
Response: Thank you for your useful suggestion, the data showed in abstrcact section was not the CAS number but the OAV values. We have revised this setence: “Among the thirty-seven aroma-active compounds, dimethyl trisulfide had the highest odor activity value (OAV) from 23300 to 87900, 3-methyl-butanal (OAV, 4568-24060), 5-ethyl-4-hydroxy-2-methyl-3(2H)-furanone (OAV, 1235-82852), and 2-methoxy-phenol (OAV, 991-2819) showed higher odor activity value.”.
The meaning of this work can provide us many directions in soy sauce production, the sentence was added at the concludion section: “In the future, the formation or derivation mechanisms of these representative aroma compounds during soy sauce fermentation or shelf life storage should be further investigated.”
- The methodology described in subchapter 2.2 "Sensory evaluation" should be checked once more. It is more likely that the first description is done for QDA and the second for CATA. Due to my knowledge, CATA is dedicated to consumer analysis, which means judges without previous training, whereas the QDA test is done mostly by trained experts. Figure 2A should be presented just below the text that described its content.
Response: At the sensory evaluation section, CATA analysis were conducted to evaluat the aroma quality of 32 kinds of soy sauce samples so as to quickly cluster the groups of soy sauce sample (Fgiure 1). Then, the sample with best comsumer’s prefernece culd be further screened from each clusters (Figure 2B). We have revised the Figures in manuscript, CATA analysis result was independantly presented in Figure 1. The QDA analysis was performed after the 7 soy sauce samples screened out from 32 samples, which presented in Figure 2C.
Figure 1. Heatmap of different brands of soy sauce samples by CATA analysis (the abbreviation of horizontal ordinate represented the name of 32 kinds of soy sauce presented in 2.1 Materials and Chemicals section).
The “2.2 Sensory evaluation” section was revised as follows:
Check-all-that-Apply (CATA) analysis. The aroma descriptions of the different soy sauce samples were analyzed by CATA analysis [18]. Soy sauce (20 mL) loaded into odorless transparent plastic bottle and coded with 3-digital numbers was submitted to the panelists randomly. The room temperature and humidity of sensory evaluation room were 24 ℃ and 55%, respectively. The procedures of soy sauce aroma profile evaluation by CATA analysis: (1) Collecting and summarizing the aroma descriptions from literatures. (2) Screening the aroma descriptors of soy sauce, and finally select 43 non repetitive sensory descriptors. (3) Statistical analyzing the aroma descriptors of 32 soy sauce samples with different description. The cluster analysis was conducted to classify the different cluster of 32 soy sauce samples. After the soy sauce clusters were determined, panelists were request to select the best acceptance samples from each cluster according to consumers’ preference [19,20].
Quantitative descriptive analysis (QDA). Twelve sensory evaluation panelists (6 males and 6 females) aged in 24-30 and with no rhinitis were recruited from our laboratory. All panelists have sensory evaluation experiences in meat products and flavorings but also familiar with the QDA evaluation methods. Panelists were trained to discriminate the aroma differences of 54-aroma kits (Le Nez du Vin, France) for three weeks before to sensory evaluate the soy sauce samples according to our previous work [21,22]. Based on the frequency statistics of aroma descriptors and the discussion of the panel group, 8 aroma attributes of soy sauce were determined: savory, caramel-like, smoky, cooked potato-like, malty, alcoholic, fruity, and sour. According to the amount of soy sauce added in the different dishes summarized from the book “A Bite of China”, then soy sauce was added into 20 mL of purified water solution. The soy sauce sample loaded in odorless transparent plastic bottle and code with 3 digits randomly, and subsequently submitted to the panelists. Panelists were required to score the intensity (1-3, weak, 4-6, medium; 7-9, strong) of 8 disused aroma attributes.
- If the calculation for volatile compounds was done only according to internal standards, it cannot be named "concentration" but "relative content".
Response: The semi-quantitative analysis was used in this work. The concentration of the aroma compounds was revised into relatve content in whole manuscript.
- What was the reason for using 2-methyl-3-heptanone as a standard? The list of standards seems not to be fulfilled.
Response: The 2-methyl-3-heptanone was not naturally present in food and could not cause the interference with other aroma compounds. It was listed at the materials and chemicals section “2-Methyl-3-heptanone (99.9%)”.
- I advise firstly to precisely present and discuss the GC-MS/O analysis and then QDA and CATA profiles. However, I have doubts about the proper CATA methodology. For sure QDA doesn't reflect consumer study. "
Response: Thanks for your suggestion. In this work, we developed a rapid screening protocol of the soy sauce for cold dishes and identification of the aroma-active compounds. Firstly, we collected 32 soy sauce samples form the market, and then we divide them into several clusters according to the aroma quality based on the CATA analysis. Finally, the specific aroma profiles of seven soy sauce samples with best consumers’ preference were screened out. Subsequently, the aroma-active compounds in seven soy sauce samples were identified by SPME-GC-MS/O.
Overall, the aim of CATA analysis was to evaluate the aroma perception of different soy sauce samples for further cluster analysis. The aim of QDA analysis was to specific the detailed aroma profiles.
Other revisions: the grammar and expression were improved by English native editor.

Reviewer 2 Report
-
-
Author Response
The grammar and expression of this manuscript were improved by English native editor.
